# Quantitative and functional interrogation of parent-of-origin allelic expression biases in the brain

Julio D Perez[1†], Nimrod D Rubinstein[1†], Daniel E Fernandez[2†], Stephen W Santoro[3†], Leigh A Needleman[1], Olivia Ho-Shing[1], John J Choi[1], Mariela Zirlinger[4], Shau-Kwaun Chen[5], Jun S Liu[2], Catherine Dulac[1]*

[1]Department of Molecular and Cellular Biology, Howard Hughes Medical Institute, Harvard University, Cambridge, United States; [2]Department of Statistics, Harvard University, Cambridge, United States; [3]Neuroscience Program, Department of Zoology and Physiology, University of Wyoming, Laramie, United States; [4]Cell Press, Cambridge, United States; [5]National Chengchi University, Tapei, Taiwan

**Abstract** The maternal and paternal genomes play different roles in mammalian brains as a result of genomic imprinting, an epigenetic regulation leading to differential expression of the parental alleles of some genes. Here we investigate genomic imprinting in the cerebellum using a newly developed Bayesian statistical model that provides unprecedented transcript-level resolution. We uncover 160 imprinted transcripts, including 41 novel and independently validated imprinted genes. Strikingly, many genes exhibit parentally biased—rather than monoallelic—expression, with different magnitudes according to age, organ, and brain region. Developmental changes in parental bias and overall gene expression are strongly correlated, suggesting combined roles in regulating gene dosage. Finally, brain-specific deletion of the paternal, but not maternal, allele of the paternally-biased *Bcl-x, (Bcl2l1)* results in loss of specific neuron types, supporting the functional significance of parental biases. These findings reveal the remarkable complexity of genomic imprinting, with important implications for understanding the normal and diseased brain.

*For correspondence: dulac@fas.harvard.edu

†These authors contributed equally to this work

## Introduction

In mammalian brains, neural computations underlying signal processing and behavioral control are conducted by a large diversity of cell types, each defined by unique but flexible patterns of connectivity, electrical properties, and profiles of gene expression and chromatin states (*Fishell and Heintz, 2013*). New experimental and conceptual frameworks have begun to address the nature of long-lasting regulatory changes at the level of the chromatin that may contribute to the establishment of neuronal identities and participate in stable encoding of cellular memories (*Dulac, 2010*).

Genomic imprinting is a unique and long lasting form of epigenetic inheritance that relies on chromatin modifications or 'imprints' established in the parental germ lines and maintained in cells of the developing and adult organism, resulting in the differential expression of the maternally- or the paternally-inherited allele (*Bartolomei and Ferguson-Smith, 2011*). Although expression differences between parental alleles are commonly assumed to be all-to-none, a substantial number of imprinted genes have been shown to exhibit a biased expression from one of the parental alleles in at least some tissues (*Dao et al., 1998*; *Lewis et al., 2004*; *Khatib, 2007*; *Babak et al., 2008*; *Menheniott et al., 2008*; *Tierling et al., 2009*; *Gregg et al., 2010*; *Gregg, 2014*). Imprinted genes have been shown to play key roles during embryonic development (*Cleaton et al., 2014*), in the placenta (*Tunster et al., 2013*), and more recently, in the developing and adult brain (*Wilkinson et al., 2007*; *Keverne, 2012*).

**eLife digest** Most cells in the human body contain two copies of each chromosome—one that was inherited from the individual's mother and one from the father—and therefore contain two copies of every gene. While both copies are usually used equally and simultaneously to produce proteins, in a minority of cases the gene from one parent is silenced in a process known as genomic imprinting. This is generally achieved via the addition of chemical marks onto the DNA, which prevent the molecular machinery that activates genes from accessing the genetic material.

Previous efforts to map imprinting in the brain throughout the mouse genome have yielded inconsistent results, due in part to the large number of factors that can affect gene expression. Perez, Rubinstein, Fernandez et al. have now addressed this issue by applying a combined approach, which includes developing a powerful statistical model that takes into account variation in age, sex, and mouse strain and extensively validating each imprinted gene candidate using an independent experimental technique.

Perez, Rubinstein, Fernandez et al. analyzed genomic imprinting initially in a part of the brain called the cerebellum in both young and adult mice. This analysis confirmed the occurrence of imprinting in 74 genes identified in previous studies, and revealed imprinting for the first time in a further 41 genes. The degree of imprinting varied between genes. In some genes only one copy was expressed and the other was completely silenced whereas others only deviated from the two copies being expressed equally. For individual genes, imprinting varied with age, tending to be more pronounced in young animals than in adults. It also varied between brain regions and typically genes were imprinted more in the brain compared to elsewhere in the body.

Mapping the activities of the imprinted genes revealed that many are involved in regulating the process of controlled cell death, or 'apoptosis'. For one particular test gene, selectively deleting either the maternal or paternal copy had different effects on the mice, thereby confirming that imprinting can affect brain development and activity. With this in mind, the potential impact of imprinting should also be considered when evaluating the effects of inherited mutations on human health.

A systematic survey of the expression patterns of known imprinted genes in the brain indicates a preferential enrichment of imprinted gene expression in a subset of brain areas, in particular those involved in feeding, social, and motivated behaviors (*Gregg et al., 2010*). Moreover, the prominent role of imprinted genes in brain development and function is evidenced by a number of human disorders and mouse mutant phenotypes (*Wilkinson et al., 2007*). For instance, patients with Prader–Willi syndrome, caused by loss of paternal expression in the q11-13 region of human chromosome 15, display abnormal development, hyperphagia, mental retardation, and volatile behavior (*Peters, 2014*). In contrast, Angelman syndrome, which is caused by loss of maternal expression in the same genomic region, results in mental retardation, impaired speech, and an abnormally joyful demeanor (*Peters, 2014*). Similarly, Birk-Barel mental retardation syndrome (*Barel et al., 2008*) is caused by mutations in the human maternally expressed *KCNK9* gene, and phenotypes associated with loss of the imprinted genes *Grb10*, *Mest*, and *Peg3* in the mouse exhibit impairments in specific social behaviors (*Lefebvre et al., 1998*; *Li et al., 1999*; *Isles et al., 2006*; *Garfield et al., 2011*).

A key feature of genomic imprinting lies in the transmission of epigenetic marks that remain stable across cell divisions throughout the lifespan of the organism and in different tissues. Surprisingly, subsets of genes have been reported to exhibit tissue-specific imprinting, and the whole brain, neurons and certain brain regions emerged as hot spots for such regulation (*Albrecht et al., 1997*; *Gregg et al., 2010*; *Sato and Stryker, 2010*; *Prickett and Oakey, 2012*). A genome-wide identification of allelic parental bias throughout the adult and developing brain appears therefore necessary to fully assess the role of genomic imprinting in the nervous system. Such a quest has been noticeably difficult to achieve. Initial methods to uncover imprinted genes based on the differential expression between parthenogenetic (containing only maternally derived chromosomes) and androgenetic (containing only paternally derived chromosomes) embryos, and subsequent discovery of adjacent imprinted loci within the genome were mainly focused on early developmental stages, and led to the identification of approximately 100 imprinted genes (*Kaneko-Ishino et al., 1995*; *Hagiwara et al., 1997*;

*Morison et al., 2005*; *Ruf et al., 2006*). The development of next generation RNA sequencing (RNA-seq) allowed for genome-wide screens of parentally biased allelic expression in any tissue of interest using F1s of reciprocal crosses between distantly related mouse strains. An intriguing question was whether or not this new, and presumably more powerful, experimental strategy would uncover novel imprinted genes. The answer to this question was proven challenging and controversial. In pioneer RNA-seq analyses of mouse hybrids, *Wang et al. (2008)* and *Babak et al. (2008)* used neonatal brains and E9.5 embryos, respectively, and determined parental biases by testing if the sum of parentally phased reads along a gene significantly deviates from biallelic expression. This approach, combined with shallow sequencing, only identified a handful of novel imprinted genes and failed to detect genes known to be imprinted in the profiled tissues. Next, *Gregg et al. (2010)* conducted an imprinting study at higher resolution by characterizing the preoptic area and prefrontal cortex of adult males and females, and the E15 brain, with an over 10-fold higher sequencing depth compared to the two previous studies. This experimental design, combined with testing for deviation from biallelic expression of parentally phased reads at each single SNP rather than along an entire gene, resulted with a much larger number of novel imprinted gene candidates. However, most of the novel imprinted candidates were not subject to independent experimental validation. In turn, *DeVeale et al. (2012)* criticized the use of single SNPs to infer imprinting and the lack of systematic independent validation of *Gregg et al. (2010)* as leading to a large and underestimated false discovery rate, and suggested that the vast majority of imprinted genes in mouse were likely already uncovered (*DeVeale et al., 2012*; *Kelsey and Bartolomei, 2012*).

These early studies indicate that several critical considerations should be taken into account to allow for accurate and powerful detection of parental bias using RNA-seq. First, the use of large numbers of biological samples is essential to achieve high statistical power; second, since transcripts are purified and sequenced it is more appropriate to quantify the relative abundances of all allele-specific transcript variants of a given gene rather than rely on the less accurate assessment of allelic expression at the gene level; third, a principled and formal statistical model should be used that explicitly accounts for the biological variability among samples and all factors in the experimental design (e.g., cross, sex, and age), and finally, a systematic independent validation of all imprinted candidates is necessary.

The present study addresses these various challenges and establishes a novel and rigorous framework for detecting imprinting from RNA-seq data. We have processed the RNA-seq data in an allele-aware manner and inferred parent-of-origin biased expression with a newly developed Bayesian regression allelic imbalance model (BRAIM), which accounts for all sources of variability in the experimental design. We further build upon these methodological and experimental advances to gain insights into the spatial and developmental dynamics of parent-of-origin allelic expression. From our study, the apoptotic pathway emerges as an important target of imprinting regulation, and we provide evidence that the paternal and maternal alleles of *Bcl-x* (*Bcl2l1*) contribute unequally to brain development. These findings open exciting new avenues for investigating epigenetic mechanisms underlying the normal and pathological brain.

## Results

### Transcriptome-wide profile of parent-of-origin expression in the mouse cerebellum

We performed a genome-wide screen for imprinted genes in the cerebellum, a brain structure involved in motor control and implicated in autism spectrum disorders (*Amaral et al., 2008*; *Tsai et al., 2012*; *Wang et al., 2014b*). The cerebellum is large enough to enable RNA profiling from tissue of a single animal and contains well-characterized cell types and connectivity. Moreover, unlike other brain structures, cerebellar development occurs largely postnatally, thus providing direct access to key neurodevelopmental processes such as cell proliferation, migration, and synaptogenesis (*Sillitoe and Joyner, 2007*; *Hashimoto and Hibi, 2012*).

The high variability of RNA expression across individuals requires the use of large numbers of biological samples to achieve statistical power and accurate detection of parental bias. Tissue was dissected from F1 mouse hybrids of Cast/EiJ and C57Bl/6J reciprocal crosses and RNA transcripts were sequenced from 48 individual cerebella representing two developmental stages and both sexes (*Figure 1A*). Half of the cerebella were collected at postnatal day 8 (P8), a period in which newly-born granule cells migrate to the inner granule layer. The remaining samples were collected from adult animals (P60). Each age group had equal numbers of males and females. Therefore, our experimental

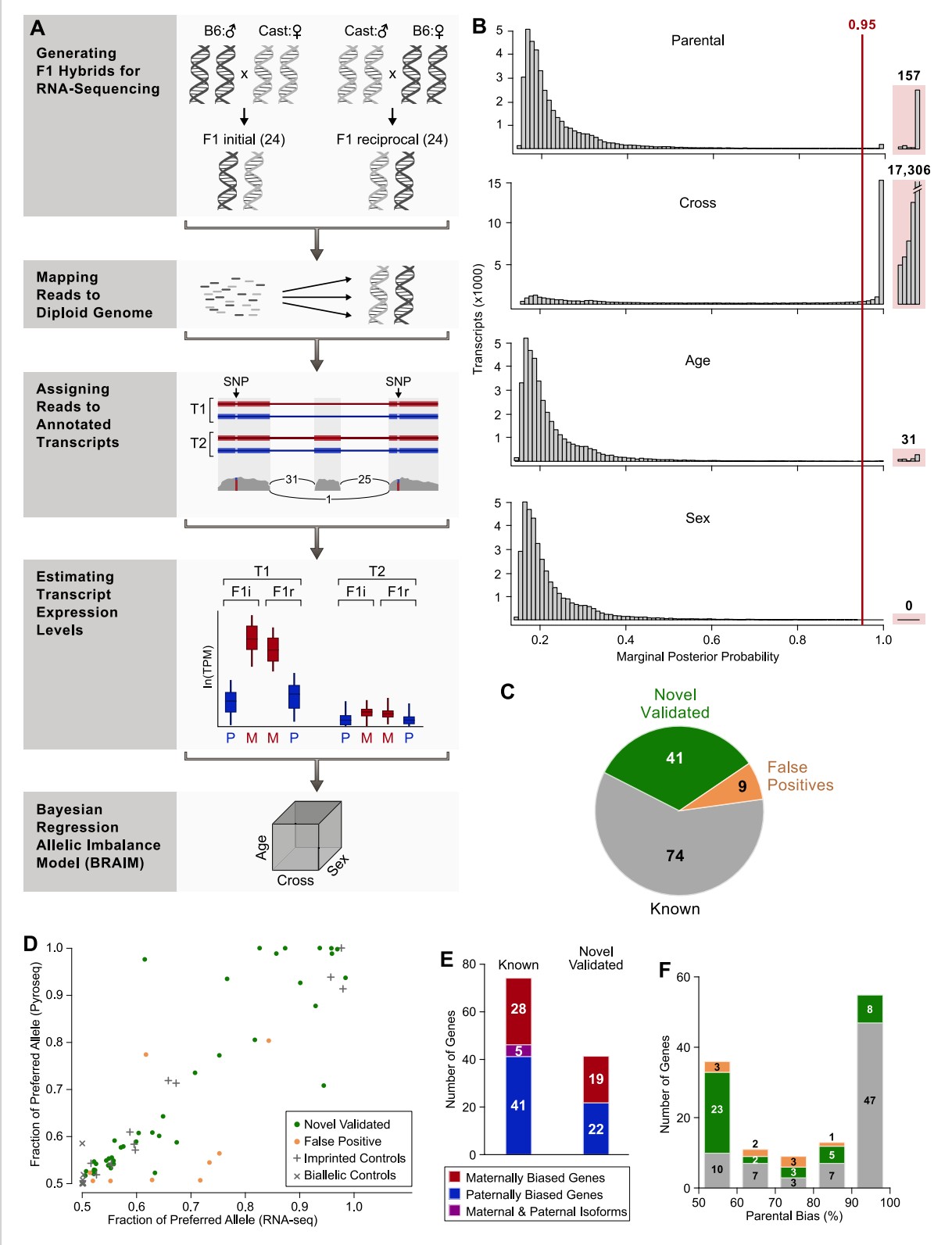

**Figure 1**. Workflow of transcriptome-wide profiling of allele-specific expression. (**A**) First, F1 hybrids were generated by crossing C57Bl/6J males with Cast/EiJ females (F1 initial) and reciprocally crossing Cast/EiJ males with females C57Bl/6J (F1 reciprocal). Second, RNA sequencing (RNA-seq) data from each of the F1 samples were mapped to a splice-junction-aware diploid C57Bl/6J, Cast/EiJ genome. Third, we transformed genomic alignments to transcriptomic alignments and filtered alignments that did not map to the transcript set using custom code. Fourth, expression levels and associated

*Figure 1. continued on next page*

*Figure 1. Continued*

errors of all expressed transcripts in the diploid C57Bl/6J, Cast/EiJ transcriptome were estimated using MMSEQ (*Turro et al., 2011*) for each mapped RNA-seq sample. Finally, for each heterozygous-expressed-transcript in the diploid C57Bl/6J, Cast/EiJ transcriptome, the parental expression bias and the effects of the mouse cross, the age, and sex were estimated using Bayesian regression allelic imbalance model (BRAIM). (**B**) Histograms of the distributions of the marginal posterior probabilities (PPs) of the parental expression biases, and the effects of cross, age, and sex of all 38,066 autosomal heterozygous transcripts in the diploid C57Bl/6J, Cast/EiJ transcriptome to which BRAIM was fitted. (**C**) Proportion of previously reported imprinted genes as well as newly identified imprinted genes subjected to pyrosequencing validation. (**D**) Relationship between pyrosequencing and RNA-seq estimates of parental biases as indicated by the percentage of expression contributed by the preferred parental allele. Note: the orange dot at ~0.8, 0.8 is considered a false positive because the preferred allele observed in the RNA-seq data is opposite to the one observed by pyrosequencing. (**E**) Number of genes exhibiting preferential expression of the maternal allele (red), paternal allele (blue), or in which the maternal and paternal alleles preferentially express different isoforms (purple) in the cerebellum. (**F**) Distribution of the magnitudes of the parental bias (% of total expression from the preferred allele) in the cerebellum.

The following figure supplement is available for figure 1:

**Figure supplement 1**. Determining expression level cutoff in RNA-seq data.

design includes six replicates for each cross, sex, and age combination, which enables inference of genomic imprinting across all individuals, while taking into account the effects of the different experimental factors.

RNA-seq libraries generated from tissues with high cellular heterogeneity and transcriptional complexity, such as found in the brain, are likely to contain genes represented by multiple transcript species (isoforms) at various abundances. Thus, accurate transcriptional analysis requires estimating the expression level of each transcript rather than quantifying expression at the gene level. In turn, quantification at the transcript level is also necessary to accurately estimate the expression of a given gene, which is merely the sum of all its isoforms (*Jiang and Wong, 2009*; *Turro et al., 2011*; *Trapnell et al., 2013*). However, since a large fraction of reads do not map uniquely to a single transcript (e.g., reads originating from constitutive exons) or even to unique genomic locations, transcript- and gene-expression levels can only be estimated with some degree of certainty, that is, with estimation error (*Jiang and Wong, 2009*; *Turro et al., 2011*; *Trapnell et al., 2013*). The uncertainty in estimating transcript- and gene-expression levels is even more pronounced when quantifying allele-specific expression, because the high sequence similarity between the two alleles results in a high level of read-mapping ambiguity. An additional concern arises from the mapping of RNA-seq hybrid data to the mouse C57Bl/6J reference genome, as it will favor the mapping of reads originating from the C57Bl/6J allele, and may therefore lead to inaccurate estimates of allele-specific expression levels (*Vijaya Satya et al., 2012*). Finally, accurate inference of genomic imprinting requires a statistical model that explicitly accounts for expression-level uncertainty in each sample, together with the biological and technical variability, and the effects of all factors in the experimental design, such as sex, age, and, in the case of mouse hybrids, the cross of each sequenced subject.

To address these challenges, we applied the following steps in the analysis of the cerebellar RNA-seq data (*Figure 1A* and 'Materials and methods'). First, we generated Cast/EiJ and C57Bl/6J diploid genomes and transcriptomes by incorporating Cast/EiJ and C57Bl/6J single nucleotide- and short insertion and deletion polymorphisms (SNPs and indels, obtained from the Mouse Genome Project: ftp://ftp-mouse.sanger.ac.uk/REL-1303-SNPs_Indels-GRCm38) into the *Mus musculus* GRCm38 reference genome sequence using the AlleleSeq package (*Rozowsky et al., 2011*). We then mapped each of the Cast/EiJ×C57Bl/6J hybrid RNA-seq libraries to the splice-junction aware diploid genome using STAR RNA-seq aligner (*Dobin et al., 2013*). For each mapped RNA-seq library, we then estimated the expression level of each transcript with its respective error, for each allele in a Cast/EiJ×C57Bl/6J diploid transcriptome using MMSEQ (*Turro et al., 2011*). Finally, we developed a statistical model testing for genomic imprinting at the level of each transcript, while taking into account the effects of cross, sex, and age across all samples. Specifically, we developed a BRAIM which is a Bayesian variable selection regression model extending *Chipman et al. (1997)* by accounting for the measurement error in the response ('Materials and methods'). In our settings, we define the response as the difference between the paternal and the maternal expression levels for a given transcript in each sample in the experiment, that is, the parental bias. The model therefore computes a posterior distribution of the

magnitude of the parental bias, and of the effects that all factors have on the parental bias, along with their marginal posterior probabilities (PPs) of being significantly different from zero.

We fitted our model to 49,464 heterozygous autosomal and X-linked transcripts (comprised of 32,399 protein-coding and 17,065 non-coding transcripts, see 'Materials and methods') from 31,547 genes expressed at levels above 0.01 transcripts per million (TPM) units (solid line in *Figure 1—figure supplement 1A* and 'Materials and methods'). A minimal threshold of 0.01 TPM was chosen because it allows for accurate and independently validated detection of parentally biased genes. We note that choosing a more stringent expression level cutoff (dashed line in *Figure 1—figure supplement 1A* and see 'Materials and methods') had a negligible effect on all the analyses described from here on. The distribution of the PPs of the parental effect for autosomal transcripts shows that most are not inferred to be imprinted (*Figure 1B* and *Supplementary file 1A–C*). It also clearly identifies a group of autosomal transcripts with PP > 0.95, which we set as our cutoff for calling an effect significant ('Materials and methods'). Our analysis also identifies a clear and substantial effect of strain on expression, as evidenced by the distribution of PPs when we account for the cross (*Figure 1B* and *Supplementary file 1A–C*). This is most likely due to the divergence of the Cast/EiJ and C57Bl/6J strains, specifically in *cis*-regulatory regions of the genes expressed in the cerebellum.

We did not identify autosomal imprinted transcripts with a sex effect with PP above the 0.95 cutoff, indicating that genomic imprinting is sex invariant in the mouse cerebellum. However, a group of autosomal imprinted transcripts were found to have an age effect with PPs above the 0.95 cutoff, indicating age-regulated imprinting (described in more detail below). Finally, among X-linked transcripts, above the PP cutoff, we detect two transcripts of the known imprinted *Xlr3b* gene (*Davies et al., 2005*), and a single transcript from each of two *Xlr3b* paralogs, *Xlr3a* and *Xlr3c* (*Supplementary file 1D–F*). The high sequence similarity between these paralogs does not allow us to distinguish which of these transcripts are actually imprinted and we therefore excluded them from the remaining analyses. Using these criteria, we identified 124 genes represented by 169 transcripts as candidate imprinted genes in our data (*Supplementary file 1G*).

Among these candidates, 74 genes were reported and validated as imprinted in previosuly published studies (*Supplementary file 1H*). The remaining 50 genes have not been previously described as imprinted (*Figure 1C*). To independently evaluate parental allelic expression bias in all these candidates, we used pyrosequencing, a real-time sequence-by-synthesis approach relying on light emissions after nucleotide incorporation (*Wang and Elbein, 2007*). As positive and negative controls, we tested 11 known imprinted genes and 11 randomly selected genes with no significant parental effects according to our RNA-seq analysis, respectively. For each candidate and control gene, we tested an average of two SNPs per gene ('Materials and methods') in 12 individual cerebella dissected from P60 and/or 12 P8 hybrids. All samples used in the pyrosequencing validations are distinct from those used in the RNA-seq experiments. We estimated parental allelic effects in the pyrosequencing data using BRAIM. The pyrosequencing data confirmed significant parental effects for 41 of the candidate novel imprinted genes and the expected significant and non-significant parental effects for all positive and negative controls, respectively (*Figure 1C,D* and *Supplementary file 1I*). Among the imprinted cerebellar genes identified, we observed a slightly greater number of genes with a paternal bias (*Figure 1E*). Interestingly, five genes preferentially express distinct isoforms from both the maternal and paternal alleles (*Figure 1E* and see below).

The distribution of the parental biases in the 115 novel and known imprinted genes spans a wide range, from a weak bias (just above 50:50%) up to strong monoallelic expression (100:0%). The bias distribution follows a bimodal shape at its two extremes (*Figure 1F*), with 36 genes displaying weak biases (from slightly above 50:50–60:40%) and 55 genes showing strong biases (90:10–100:0%). The remaining 32 genes show a moderate bias, from above 60:40% to below 90:10%. Interestingly, the weak bias mode is enriched with newly identified imprinted genes (yet includes 10 detected known imprinted genes) whereas stronger biases are enriched with known imprinted genes (yet includes 18 newly detected imprinted genes). A strong correspondence (Pearson correlation coefficient = 0.91, p-value < $10^{-16}$) is found between the biases identified by RNA-seq and pyrosequencing, respectively, thus suggesting high accuracy in their quantification (*Figure 1D*).

*Figure 2* shows examples of various levels of allelic parental biases. A newly detected imprinted gene, *Fkbp6*, shows almost exclusive expression from the paternal allele, while *Ago2* and *Nhlrc1* show moderate to weak parental biases, respectively, and *Clptm1l* is accurately detected as biallelically-expressed. Importantly, these effects appear highly reproducible across cerebellum samples analyzed both with

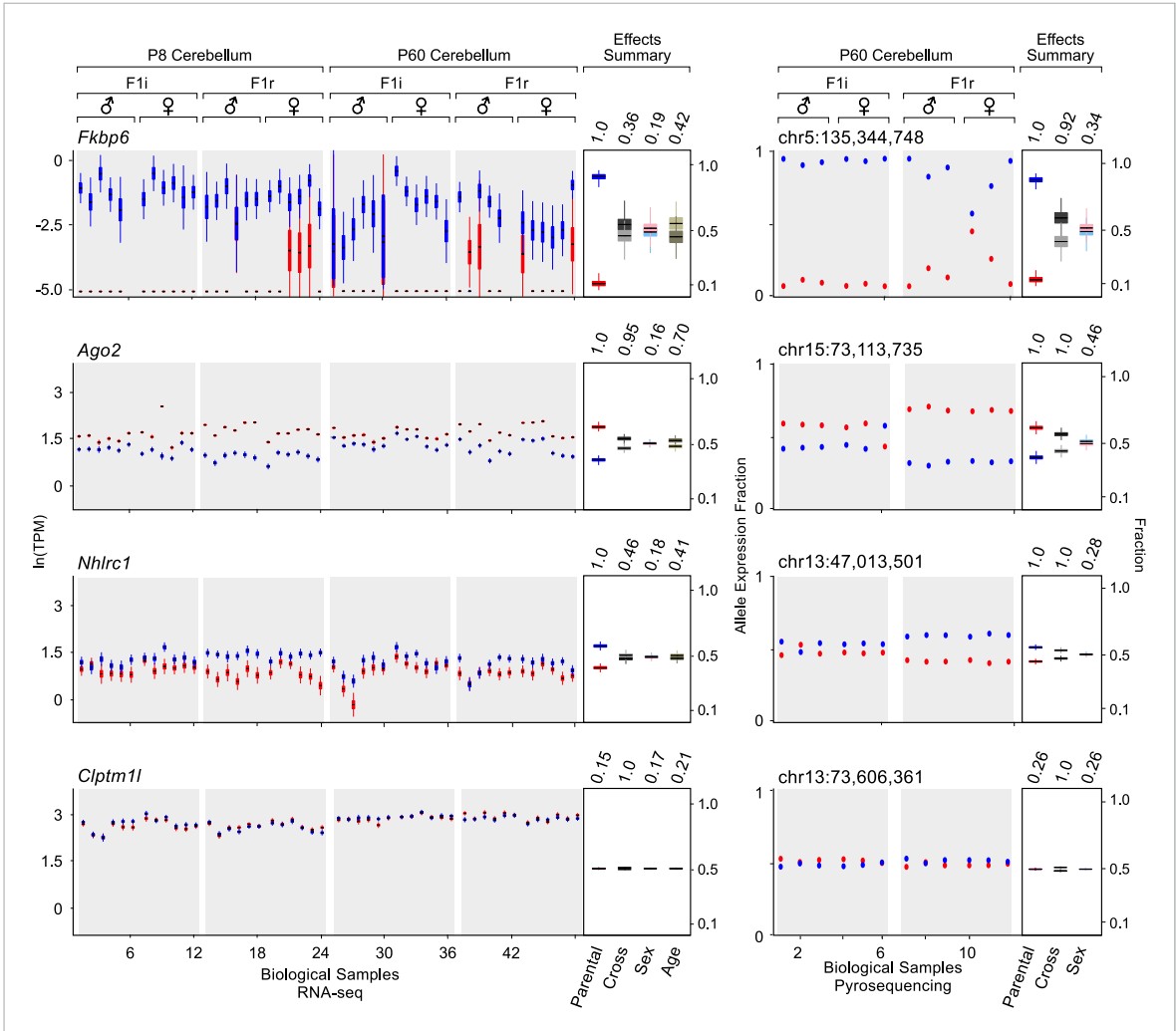

**Figure 2**. Examples of genes imprinted in the cerebellum and of a biallelic control. *Fkbp6*, *Ago2*, and *Nhlrc1* show parentally-biased expression in the cerebellum as observed by RNA-seq (left) and confirmed with pyrosequencing (right). *Clptm1l* shows biallelic expression in the cerebellum in both RNA-seq and pyrosequencing experiments. For each replicate (N = 48), red indicates maternal expression while blue indicates paternal expression. The Y-axis shows the RNA-seq expression level in natural log of TPM units (ln(TPM)), as derived from the posterior distribution of expression levels reported by MMSEQ. Each box is centered at the posterior mean, extends one posterior standard deviation away, and the bottom and top notches are the minimum and maximum posterior samples, respectively. Effects summary shows the posterior distributions of the effects of all experimental factors: parental, cross, sex, and age (box centered at the posterior mean, extends one posterior standard deviation away, and bottom and top notches are the minimum and maximum posterior samples, respectively) with their respective PPs on top. For the parental effect, blue represents paternal and red represents maternal expression. For the cross effect, gray represents F1r and black represents F1i. For the sex effect, pink represents female and cyan represents male. For the age effect, light khaki represents P8 and dark khaki represents P60.

RNA-seq and pyrosequencing. Considering 74 known imprinted genes detected in this analysis as true positives, together with 41 newly validated imprinting genes out of 50 novel candidates (altogether represented by 160 transcripts), the precision of our approach is ~93% and increases the number of identified mouse imprinted genes by ~30% (from 138 to 179, see *Supplementary file 1G, H*).

We considered the possibility that true imprinted genes may not meet our parental bias PP cutoff (i.e., false negatives). We therefore handpicked 18 genes below our 0.95 cutoff that displayed a trend resembling the allele-specific expression patterns of weakly imprinted genes. Using pyrosequencing we successfully confirmed a significant parental effect in 10 of these 18 genes (*Supplementary file 1I*). For example, *Casd1*, a gene reported to be imprinted in other tissues (*Ono et al., 2003*), only displayed a paternal effect with PP of 0.89 in the RNA-seq analysis, but showed a mild paternal bias

(52:48%; PP = 0.95) by pyrosequencing. This observation supports the existence of additional parental biases in our data that do not meet the chosen 0.95 cutoff.

## Isoform-specific imprinting

Our approach readily estimates parental bias at the transcript level, as seen in the allelic analysis of the 11 expressed isoforms of the imprinted gene *Rian* (*Hatada et al., 2001*) (*Figure 3A*). Our data suggest that most imprinted genes, including *Rian*, show a consistent imprinting pattern in all isoforms. However, isoform-specific imprinting is readily identified by our approach, and can be seen in two different genomic contexts. A few genes harboring a paternally expressed gene inserted within an intron (often a retrogene) have been shown to generate different isoforms from the maternal and paternal alleles (*Wood et al., 2008*; *Gregg et al., 2010*; *Cowley et al., 2012*). Although it is unclear exactly how such regulation arises, transcriptional interference by the intronic paternally expressed gene may play an important role (*McCole and Oakey, 2008*). Our analysis readily identified previously reported cases of genes in which isoforms are subject to such regulation (e.g., the *H13-Mcts2* locus, *Figure 3B*). Moreover, it detected additional imprinted transcripts in the *Herc3* gene- either or both of two short transcripts (indistinguishable by our sequence data) from a promoter upstream to the large 25 exons-long transcript (Gencode transcript IDs: ENSMUST00000141600.1 and ENSMUST00000122981.1), which are preferentially expressed by the maternal allele (*Figure 3C*). Other known cases of isoform-specific imprinting are due to differential methylation of alternative promoters (*Arnaud et al., 2003*; *Choi et al., 2005*; *Peters and Williamson, 2007*). Interestingly, we detect a novel maternally expressed short transcript (791 bp) (Gencode transcript ID: ENSMUST00000149496.1) at the locus of the paternally expressed *Mest* gene, whose transcription starts at exon 9, suggesting alternative promoter usage (*Figure 3D*). This transcript is presumably a non-coding RNA since no open reading frame could be identified. In total, we detected 8 of the 10 known cases of isoform-specific imprinting (the missing two cases are the short isoform of *Cdh15*, which is not expressed in the cerebellum, and the long isoform of *Blcap*, which is not heterozygous in the Cast/EiJ×C57Bl/6J hybrids), and we uncovered additional imprinted isoforms, including a novel example of isoform-specific imprinting in the *Mest* locus. For a complete list of genes exhibiting isoform-specific imprinting see *Supplementary file 1G*.

## Developmental regulation of genomic imprinting in the cerebellum

Comparison of imprinted gene expression in the cerebellum in adulthood and at P8, a critical milestone of cerebellar development during which granule cells undergo cell division, migration, and synaptogenesis (*Sillitoe and Joyner, 2007*; *Hashimoto and Hibi, 2012*) shows that the imprinted status of eight genes expressed at both P8 and P60 changes according to age (7 genes are exclusively imprinted at P8 and one is exclusively imprinted at P60 *Figure 4A*). In addition, we observe changes in the magnitude of the parental bias according to age for 29 out of 107 imprinted genes that are expressed at both age groups, with the majority exhibiting a stronger parental bias at P8 than in the adult (*Figure 4B* and *Supplementary file 1A*) Moreover, analysis of the overall expression level (sum of paternal and maternal allelic expression) reveals that 34 of the 107 imprinted genes are differentially expressed between P8 and P60 (*Supplementary file 1A*), together with 8 imprinted genes that are exclusively expressed at P8. Similar to the parental bias, the majority of the 34 age-regulated genes show higher expression in the developing cerebellum than in the adult (*Figure 4C*). Altogether, 59 imprinted genes (51% of all imprinted genes expressed in the cerebellum) are regulated in either parental bias and/or overall expression level according to developmental stage (age effect PP > 0.95). For the majority of these genes, the highest magnitude of the parental bias and the highest level of expression are observed at P8, suggesting a potential role for these genes in neurodevelopmental processes.

Next, we examined the relationship between changes in the parental bias of a transcript and changes in its expression level. Our analysis shows that the age effects on the magnitude of the parental bias and on the overall expression are positively correlated, such that both the magnitude of the parental bias and the overall expression level of imprinted genes tend to increase or decrease together with age (Pearson correlation coefficient = 0.33; p-value = $4 \times 10^{-4}$, *Figure 4D*). Since the ability to accurately detect and estimate parental biases is limited at low expression levels, we repeated this analysis with a higher expression level threshold (right to the dashed line in *Figure 1—figure supplement 1A,B*) but found no significant effect on our results. The relationship between parental bias and overall expression identified in our analysis may result from the fact that the two parental alleles experience differential

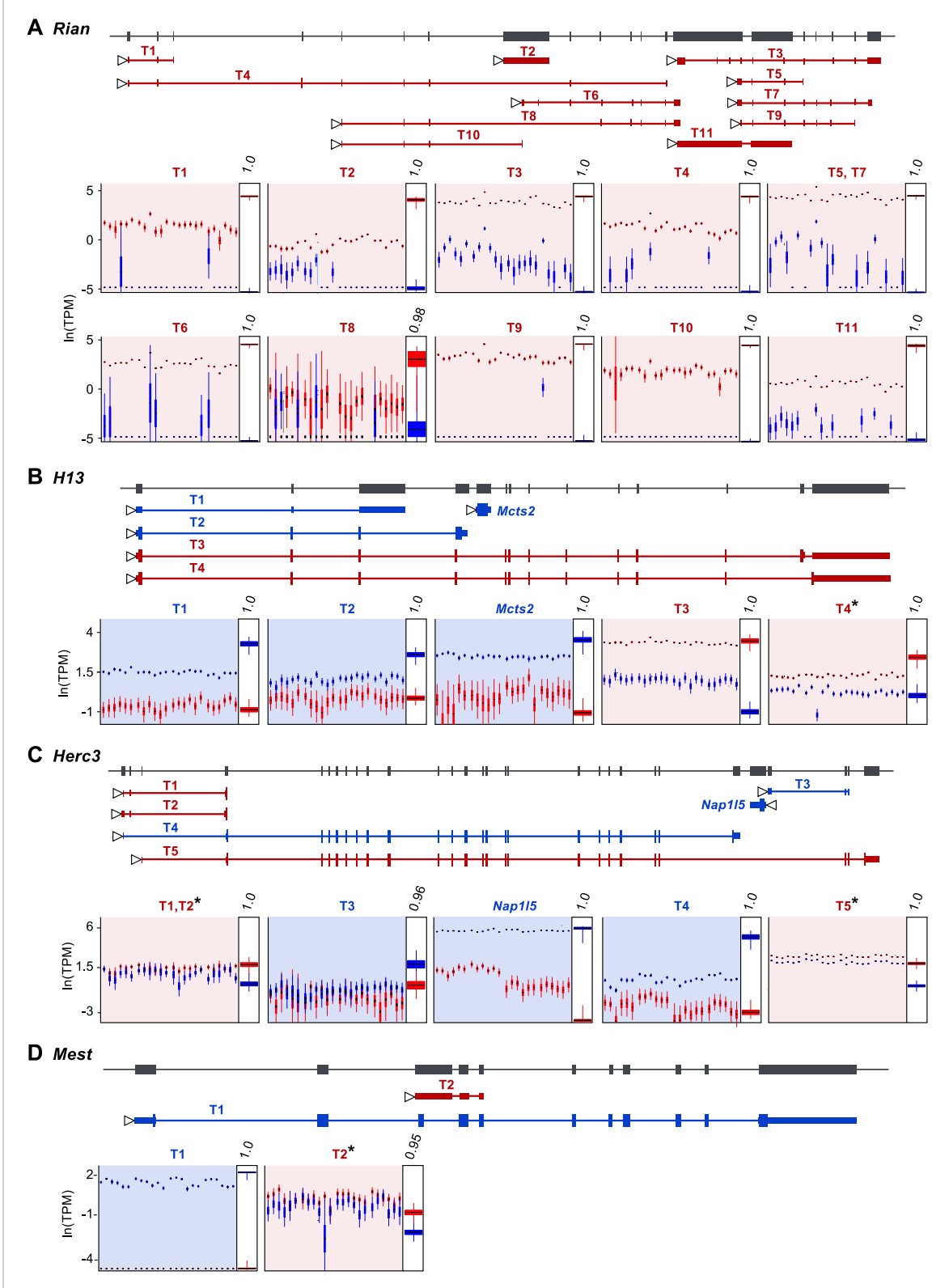

**Figure 3**. Assessment of imprinting at the transcript level. (**A**) Imprinting in all identified spliced variants of *Rian* in the P8 cerebellum. (**B**) Imprinting of *Mcts2* and *H13* transcripts in the P8 cerebellum. (**C**) Imprinting of *Nap1l5* and *Herc3* transcripts in the P60 cerebellum. (**D**) Imprinting in transcripts of *Mest* in the P60 cerebellum. In all figures, gene models are in dark gray and transcript models are colored according to the preferentially expressed parental allele (maternal in red and paternal in blue). Parental-specific expression across all replicates is shown below transcript models in natural log of TPM units *Figure 3. continued on next page*

Figure 3. Continued

(ln(TPM)), with the posterior distribution of the parental biases across replicates indicated in the right inset boxes with corresponding PPs on top. Asterisks indicate that the associated transcript is developmentally regulated.

transcriptional regulation during development, thereby altering both the magnitude of the parental bias as well as the overall level of expression. To test this hypothesis we fitted our model to the data where we defined the response as either the paternal or the maternal expression level. This analysis reveals that age regulated expression or parental bias (age effect PP > 0.95) is achieved either by a significant change in the expression level only of the most highly expressed (preferred) parental allele (PA; 19 genes, aligned along the X-axis in *Figure 4E*), a significant change in the expression level only of the non-preferred allele (8 genes, aligned along the Y-axis in *Figure 4E*), or a significant change in the expression levels of both alleles (12 genes, aligned along the diagonal in *Figure 4E*). Finally, some genes have a significant change in expression and/or parental bias only when both alleles are combined but not when the alleles are analyzed separately (11 genes, scattered around the origin of axes in *Figure 4E*). Thus, our data suggest that altering the expression level of the preferred allele is the most common mode through which imprinted genes are regulated according to age (p-value = 0.03; $\chi^2$ test), and this allelic regulation changes both the parental bias as well as the overall expression level. It is also possible that the enhanced expression and parental bias observed in the P8 cerebellum originates from different cellular compositions of the developing and adult cerebellum. These scenarios are not mutually exclusive.

A number of genes for which the parental bias in the cerebellum is affected by age are associated with developmental processes such as cell proliferation, differentiation, and survival. For instance, the *Asb4* gene, which regulates embryonic stem-cell differentiation (*Townley-Tilson et al., 2014*), exhibits highly maternally-biased expression during cerebellar development but is biallelically expressed during adulthood, which is achieved by a significant decrease in maternal expression and a slight increase in paternal expression (*Figure 4F*). The growth suppressor *Grb10* gene exhibits biallelic expression at P8 but exclusive paternal expression in the adult resulting from silencing of the maternal allele (*Figure 4F*). Interestingly, we observed a switch in the parental bias for the transcription factor *Zim1*, from maternal during development to paternal in the adult cerebellum due to a reduction in maternal expression level (PP = 1.0), and no change in paternal expression (PP = 0.19) (*Figure 4—figure supplement 1*).

Our analysis uncovered a novel imprinted locus at the distal end of chromosome 1, which exhibits age-dependent regulation. The genes *Ier5*, *Mr1*, *Stx6*, and the putative *BC034090* gene, which we name here *Impar* (for Imprinted and Age Regulated) are located side by side within 136 KB and show a maternal bias during cerebellum development but biallelic expression in the adult (*Figure 4F,G*). For all genes in this locus, the shift in parental bias is accompanied by a reduction in the expression level of the maternal allele and to a lesser extent in the paternal allele (*Figure 4F* and *Figure 4—figure supplement 1*). *Stx6* has been shown to regulate neuronal migration and the formation of neural processes (*Kabayama et al., 2008*; *Tiwari et al., 2011*), two events necessary for the integration of granule cells to the cerebellar circuit occurring at the P8 stage. It would therefore be interesting to determine whether the allelic regulation of this gene, which directly affects its expression level, is critical for this process. Finally, we also observe age effects on the parental biases of specific isoforms of *Herc3*, *Mest*, and *H13*, which all show isoform-specific imprinting (asterisks in *Figure 3B–D* and *Figure 4—figure supplement 2*).

## Genomic localization

Of the 115 genes found imprinted in the cerebellum, 106 (92%) are located within 1 MB of at least one other imprinted gene. The remaining 9 genes (8%) are isolated from other known or newly identified imprinted genes. Moreover, although most newly identified imprinted genes from this study are located within or around known imprinted clusters, 7 of the 41 (17%) appear isolated in the genome (*Figure 5A,B*). We noticed that a substantial number of imprinted genes displaying weak to moderate parental bias localize to the vicinity of imprinted genes showing stronger parental bias (*Figure 5A*). For example, *Ankrd34c* and *Ctsh*, two genes that exhibit a weak paternal bias (note: the parental bias of *Ankrd34c* is significant according to the pyrosequencing data only), are located up- and downstream to *Rasgrf1*, a gene exclusively expressed from the paternal allele (*Figure 5C*). This observation led us to explore

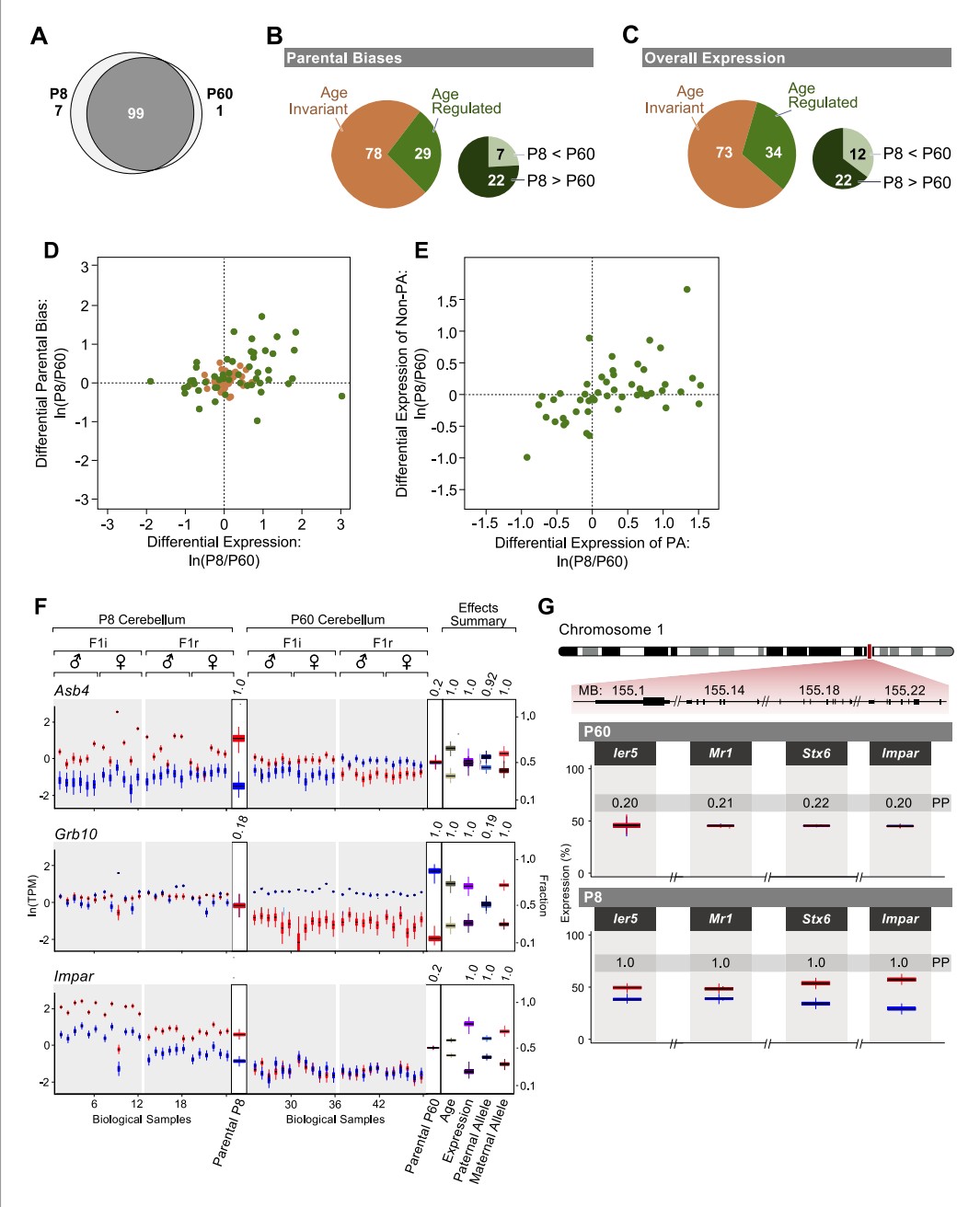

**Figure 4**. Age regulation of imprinted genes in the cerebellum. (**A**) Venn diagram of imprinted genes at P8 and at P60. (**B**) Proportions of imprinted genes in which the parental bias is regulated during cerebellum development. Age regulated genes (in green) include genes with significantly higher parental bias at P8 (dark green) and genes with significantly higher parental bias at P60 (light green). (**C**) Proportions of imprinted genes with overall expression regulated according to the developmental stage of the cerebellum. Age regulated genes (in green) include genes with significantly higher overall expression at P8 (dark green) and genes with significantly higher overall expression at P60 (light green). (**D**) Relation between age fold change (ln(P8/P60)) of the overall expression level of imprinted genes and their parental bias. Genes located in the top right quadrant exhibit higher overall expression and stronger parental biases at P8 than at P60. Genes located in the lower left quadrant exhibit lower overall expression and lower parental biases at P8 than at P60. Genes located in the top left quadrant exhibit lower overall expression but stronger parental biases at P8 than at P60. Genes located in the bottom right quadrant exhibit stronger expression but lower parental biases at P8 than at P60. Although the age-invariant genes (salmon colored circles) overlap to some extent with the age-regulated genes (green colored circles), their age PPs are lower than the 0.95 cutoff. (**E**) Relation between age fold change (ln(P8/P60)) of the expression of the preferred (PA) and the

*Figure 4. continued on next page*

*Figure 4. Continued*

non-preferred allele (Non-PA) of imprinted genes. Genes aligned along the X-axis display age regulated expression of the PA, while genes aligned along the Y-axis show age regulated expression of the non-PA. Points aligned on the diagonal are imprinted genes with age-regulated expression of both alleles. Genes in which a significant age effect on parental bias and/or overall expression requires change in both alleles are displayed by points scattered around the origin of axes. (F) Three examples of age-regulated imprinted genes. For each replicate (N = 48), maternal expression is in red and paternal expression is in blue. Y-axis is the RNA-seq expression level in natural log of TPM units (ln(TPM)). Effects summary shows the posterior distributions of the effects of the experimental factors with their respective PPs on top. For the age effect, light khaki represents P8 and dark khaki represents P60. Expression effect shows the fraction of overall expression at P8 and P60 in magenta and purple, respectively. Paternal-allele effect shows the fraction of expression exclusively form the paternal allele at P8 and P60 in light blue and royal blue, respectively. Maternal-allele effect shows the fraction of expression exclusively form the maternal allele at P8 and P60 in light red and dark red, respectively. (G) A novel imprinted cluster at distal chromosome 1 exhibits age-dependent regulation of parental bias (corresponding PPs of the parental biases are indicated).

The following figure supplements are available for figure 4:

**Figure supplement 1**. Detection of changes in both parentally biased expression and overall expression levels.

**Figure supplement 2**. Developmental regulation of isoform-specific imprinting and/or expression.

whether parental bias consistently decays as a function of the distance from strongly biased genes. We first defined imprinted cluster centers as genes for which the parental bias of the preferentially expressed allele reaches at least 85%, and subsequently assigned adjacent imprinted genes with an intergenic distance of up to 1 MB as members of the same cluster ('Materials and methods'). This step partitioned the imprinted genes in our data to a total of 24 imprinted clusters. Analysis of the magnitude of the parental bias as a function of the distance to the cluster center reveals a statistically significant negative effect (p-value = $1.08 \times 10^{-4}$, 'Materials and methods'), supporting a model of decay of parental bias from imprinted cluster centers (*Figure 5D*). Importantly, not all of the imprinted clusters follow these patterns. For instance, the developmentally regulated imprinted cluster on chromosome 1 (described above and in *Figure 4G*) does not contain any strongly imprinted gene that could be categorized as a cluster center, nor does it exhibit a decay in the parental bias of adjacent genes.

The clustering of imprinted genes is regarded as a hallmark of their genomic organization and is thought to reflect common regional control within a given cluster (*Reik and Walter, 2001*). Our observations suggest that genes with weak to moderate parental biases may be indirectly affected by the silencing taking place at adjacent and strongly biased imprinting cluster center genes, perhaps through local chromatin conformation. Alternatively, more directed processes may regulate weak to moderate parental biases as these biases may be functionally important. Moreover, genes with weak biases in some tissue may display robust parental biases in specific cell types or developmental stages.

We reasoned that if the imprinting regulation of genes in which we detect weak-to-moderate parental bias at the boundaries of imprinted clusters is functionally important, natural selection may operate against disruption of the clustered organization, and thus the clustered organization, or micro-synteny, of such genes would tend to be conserved during mammalian evolution. Alternatively, if imprinting regulation of genes with weak-to-moderate parental bias is not functionally important, and simply reflects the indirect byproduct of regulation of adjacent genes, the micro-synteny of these genes is not expected to be strongly conserved across mammalian genomes. To test these hypotheses, we derived all pairs of adjacent genes in the mouse genome with orthologs in at least one of 15 other mammalian genomes (409,874 gene pairs) and estimated the tendency of their mammalian orthologs to be adjacent as well (*Figure 5E,F*, and 'Materials and methods'), using a probabilistic phylogenetic model analyzing phyletic patterns of presence and absence (*Cohen et al., 2008*). This analysis reveals that the mean propensity of adjacency of mammalian orthologs of weakly imprinted gene pairs in the mouse (parental biases below 85%) is significantly higher than the mean propensity of adjacency of orthologs of all mouse adjacent gene pairs (p-value = 0.007, *Figure 5F*). These findings suggest that

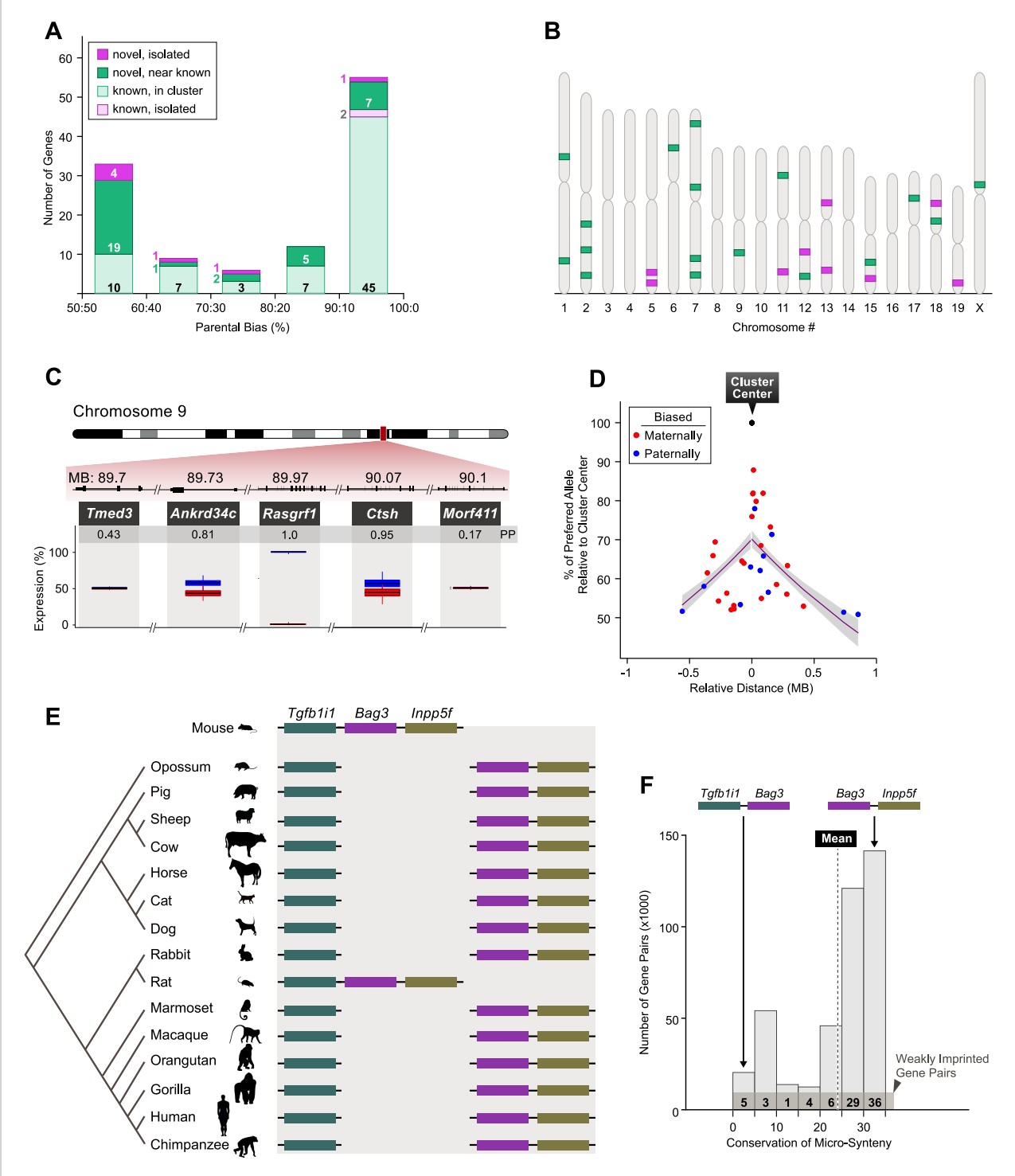

**Figure 5.** Genomic patterns of mild to moderate parental biases. (**A**) Distribution of imprinted genes according to the level of parental bias and genomic location relative to the nearest imprinted gene. (**B**) Chromosomal locations of novel imprinted genes (green: novel imprinted genes within 1 MB of known ones, purple: isolated novel imprinted genes). (**C**) Example of an imprinted cluster on chromosome 9 where the monoallelically expressed imprinted gene *Rasgfr1* is flanked by newly identified imprinted genes exhibiting moderate parental biases. (**D**) The decay of parental bias, as a function of the distance from an imprinted cluster center. Magenta: regression line, gray polygon: corresponding standard errors. (**E**) Example of mammalian conservation of the micro-synteny of the *Tgfb1i1*, *Bag3*, and *Inpp5f* genes within an imprinted cluster on distal chromosome 7. (**F**) Distribution of the mammalian conservation of micro-synteny of mouse gene-pair orthologs. The numbers of weakly imprinted gene-pair orthologs are indicated at the bottom and the vertical dashed line indicates the mean conservation across all pairs. (See 'Materials and methods' for further details.)

the micro-synteny of imprinted clusters is evolutionary conserved, thus supporting the idea that the imprinting regulation of mouse genes with weak-to-moderate parental biases is functionally important. This also suggests that the orthologs of many of these genes may be imprinted in other mammalian species.

In addition to newly identified imprinted genes associated with known imprinted clusters, several novel imprinted genes appear isolated from any other known imprinted gene (>2 MB away). Previous studies have reported a differential methylation between the two parental alleles for some of these genes. For example, differential methylation was observed in a region of chromosome 13 immediately downstream of *Nhlrc1* (*Xie et al., 2012*), a novel paternally-biased gene in our results (*Figure 2*). In humans, mutations in *Nhlrc1* cause Lafora progressive myoclonic epilepsy (*Romá-Mateo et al., 2012*), a fatal neurological disorder characterized by the presence of massive intracellular inclusions observed in several neuronal cell types across the brain including the cerebellar granule cells. Differential methylation but not parentally biased expression in the brain was also reported within the Actinin alpha 1 (*Actn1*) gene at chromosome 12 (*Calaway and Domínguez, 2012*), which codes for a protein that regulates cytoskeleton interactions with the membrane. Our results show that this gene is indeed preferentially expressed from the paternal allele in the cerebellum. We also uncovered the paternal-specific expression of the *Fkbp6* gene, which is isolated on distal chromosome 5 (*Figure 2*). Interestingly, this gene displays maternal allele-specific binding of ZFP57, a DNA-binding protein that specifically binds to the majority of imprinted genes and protects them from demethylation after fertilization (*Quenneville et al., 2011*).

## Spatial regulation of genomic imprinting

Tissue-dependent regulation of imprinting has been described for a subset of known imprinted genes (*Gregg et al., 2010*; *Prickett and Oakey, 2012*). We selected 28 imprinted genes (20 known and 8 novel representing multiple independent clusters) based on the most interesting predicted biological functions and performed a systematic pyrosequencing quantification of the parental expression bias in 16 brain macro-regions and seven non-neural peripheral tissues in the adult (*Figure 6A,B* and 'Materials and methods'). This analysis revealed pronounced changes of parental bias across genes and tissues (*Figure 6C,D* and *Supplementary file 1J, K*).

We found that a number of previously known imprinted genes such as *Igf2*, *Igf2r*, *Zim1*, *Copg2*, and *Ago2* display noticeable differences either in their imprinting status and/or direction of parental bias across different brain regions and in non-brain tissues (*Figure 6C,D* and *Supplementary file 1J, K*). In particular, the significant differences observed between brain regions and non-neural tissues suggest different contributions from the parental genomes to these regions of the body. *Igf2*, previously shown to be maternally biased in the adult cortex and preoptic area (*Gregg et al., 2010*), is indeed confirmed to be robustly maternally biased in all 16 brain regions tested, while it is mainly expressed from the paternal allele in non-brain tissues. This stands in contrast to *Grb10*, which is paternally biased in the brain and maternally biased in the body (*Charalambous et al., 2003*, *2010*; *Garfield et al., 2011*) (*Figure 6D*). *Igf2r*, which exerts a function that is antagonistic to that of the paternally expressed *Igf2*, has been shown to be maternally expressed during development (*Filson et al., 1993*; *Ludwig et al., 1996*) and in most adult tissues with the exception of the brain (*Hu et al., 1999*). This pattern is also observed in our data (*Figure 6C,D*). Finally, the putative transcription factor *Zim1* shows an intriguing pattern of both paternal and maternal biases both in the brain and in the body (*Figure 6C,D*).

Hierarchical clustering of the parental biases of the tested genes showed three main clades comprising maternally biased genes, paternally biased genes, and genes that are sporadically biased in the brain. Several genes exhibit sharp contrasts in parental bias between the brain and body. This includes genes that are exclusively or nearly exclusively biased in and throughout the brain, such as the maternally-biased Ube3a (*Rougeulle et al., 1997*), *Trappc9*, *Bag3*, and *B3gnt2* genes and the paternally-biased *Bcl-x* long isoform (*Bcl-x$_L$*), *Inpp5f_v2* isoform (*Choi et al., 2005*), and *Begain* gene (*Figure 6C,D*).

Hierarchical clustering also clearly separates tissues into two main clades, non-brain and brain, with parental bias in brain appearing much more consistent and robust. The brain is further subdivided into additional sub-clades, which roughly group developmentally related regions. One sub-clade clusters

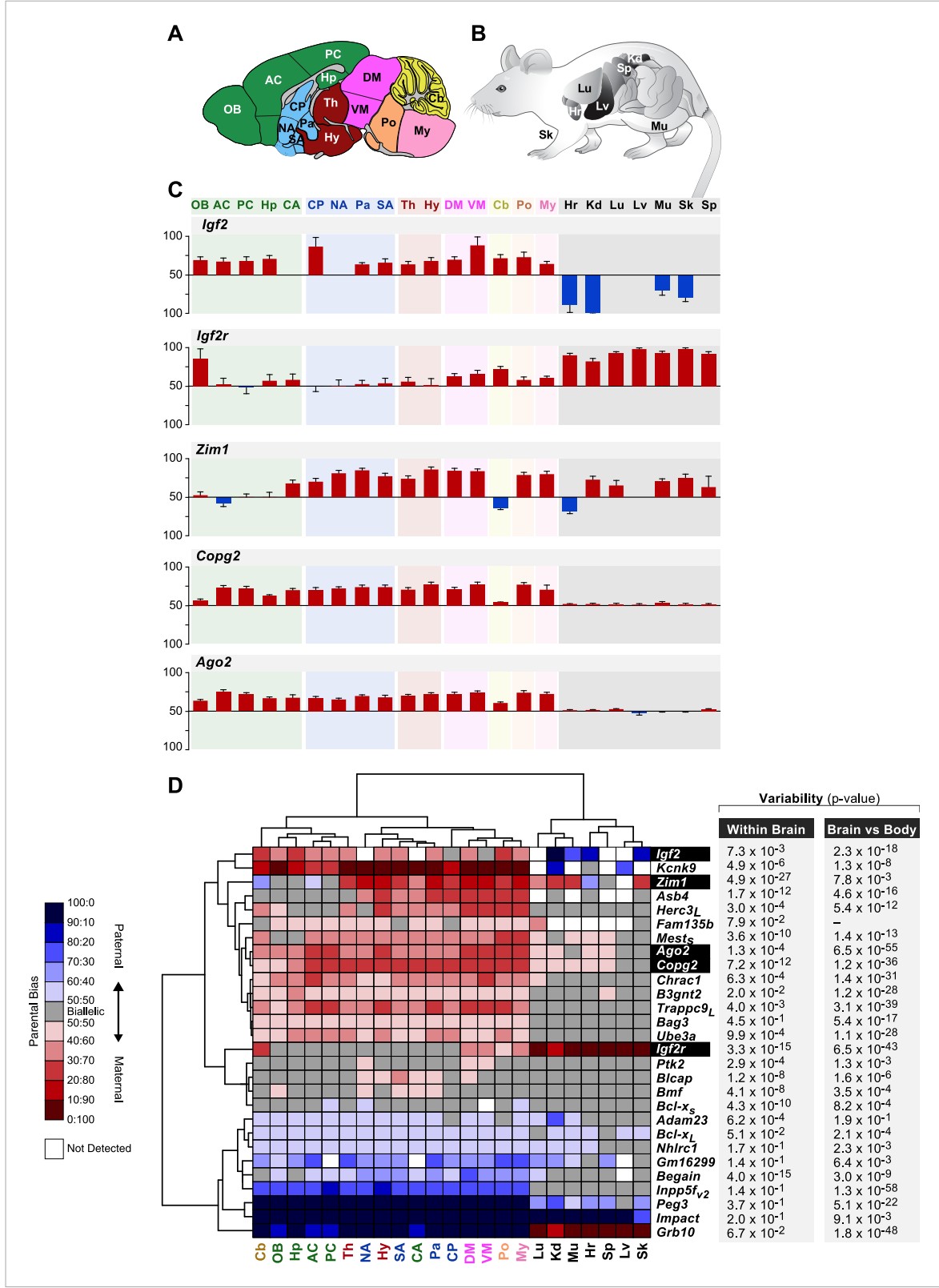

**Figure 6.** Spatial regulation of imprinted genes. (**A**) Legend of brain regions analyzed, colored according to their broad developmental relatedness. OB: Olfactory Bulb, AC: Anterior Cortex, PC: Posterior Cortex, Hp: Hippocampus, CA: Cortical Amygdala (which is lateral to the brain midline and hence not captured by this sagittal section), CP: Caudate Putamen, NA: Nucleus Accumbens, Pa: Pallidum, SA: Striatum-like Amygdala, Th: Thalamus, Hy: Hypothalamus, DM: Dorsal Midbrain, VM: Ventral Midbrain, Cb: Cerebellum, Po: Pons, My: Medulla. (**B**) Legend of body tissues analyzed. Lg: Lung,

*Figure 6. continued on next page*

*Figure 6. Continued*

Hr: Hearth, Sp: Spleen, Lv: Liver, Kd: Kidney, Sk: Skin, Mu: Muscle. (**C**) Examples of genes whose parental bias is regulated according to organ (brain vs body tissues) and of genes whose parental bias is dynamically regulated across the brain. Origins of bar graphs represent biallelic expression. Positive values represent preferential maternal expression (colored red) while negative values represent preferential expression of the paternal allele (colored blue). N = 6 in each bar. (**D**) Hierarchical clustering of the heat map representing deviations from biallelic expression (N = 6 in each square) in imprinted genes. To the right of the heat map are the ANOVA p-values testing for variability of parental bias across the brain regions (left columns) and paired *t*-test p-values testing for differential parental bias in brain vs body (right column).

The following figure supplements are available for figure 6:

**Figure supplement 1**. Hierarchical clustering of the heat map representing deviations from biallelic expression of imprinted genes in the brain.

**Figure supplement 2**. Shared patterns of spatial regulation of parental biases for different imprinted genes in the brain.

most of the telencephalon, the thalamus, and cerebellum, another groups mesencephalic and rhombencephalic regions, and a third groups diencephalic and basal ganglia regions (*Figure 6D*).

In order to further identify common patterns of parental bias across the brain, we performed a hierarchical clustering analysis confined to the 16 brain regions surveyed, excluding peripheral tissues (*Figure 6—figure supplement 1*). This analysis revealed that *Ago2*, *Chrac1*, and the long isoform of *Trappc9*, which co-localize to an imprinted cluster in the distal end of chromosome 15, exhibit a similar pattern of maternal bias across the brain, stronger in the cortex and weaker in the olfactory bulb, hippocampus, and cerebellum (*Figure 6C* and *Figure 6—figure supplement 2A* and *Supplementary file 1K*). Interestingly, *Copg2* and the short isoform of *Mest*, which co-localize near the centromeric region of chromosome 6, exhibit a similar pattern of bias to that of genes on the distal end of chromosome 15 (*Figure 6C* and *Figure 6—figure supplement 2A* and *Supplementary file 1K*). The *Zim1*, *Asb4* genes, and *Herc3* long isoform, which are located in the proximal end of chromosome 7, the proximal end of chromosome 6, and near the centromeric region of chromosome 6, respectively, also exhibit shared patterns of biallelic expression (or in the case of *Zim1* a weak paternal bias) in telencephalic regions and cerebellum but strong maternal biases in other brain regions (*Figure 6—figure supplement 2B* and *Supplementary file 1K*). These results suggest that the brain executes region-specific programs of imprinting involving genes from the same or distinct imprinted clusters.

## Genomic imprinting in the developing brain

The differences in parental biases observed across the brain raise the question of when during development these specificities are established. To address this issue, we performed a pyrosequencing analysis of the parental biases of 13 genes inferred to be temporally and/or spatially regulated across the brain and in non-brain tissues, at postnatal days 0, 8, 15, and 64 in the cortex, hypothalamus, and cerebellum (*Supplementary file 1I*). These three brain regions were selected for developmental analysis because they display contrasting imprinting patterns in the adult. In addition, we analyzed the parental bias of these 13 genes in the entire E15 brain. This analysis revealed substantial spatiotemporal dynamics of the parental bias of several genes (*Figure 7A,B*). For instance, *Blcap* exhibits a gradual decrease in maternal bias over development consistently across the three analyzed brain regions. In contrast, the switch from maternal to paternal bias of *Zim1* during cerebellum development, occurs gradually along development in both the cerebellum and cortex, yet is not mirrored in the hypothalamus where maternal bias is strongly maintained into adulthood. Moreover, this analysis reveals that the sharp contrast between the parental biases of *Igf2* and *Grb10*, observed across the brain, seems to be temporally co-regulated. The switch in the expressed allele for both genes happens earlier in the cortex and hypothalamus than in the cerebellum, which roughly coincides with the completion of their development (*Levitt et al., 1997*; *Sillitoe and Joyner, 2007*; *Shimogori et al., 2010*). These results demonstrate that imprinting is a remarkably dynamic process, and suggest that the highly coordinated spatiotemporal regulation of parent-of-origin expression may in turn orchestrate development across different brain regions.

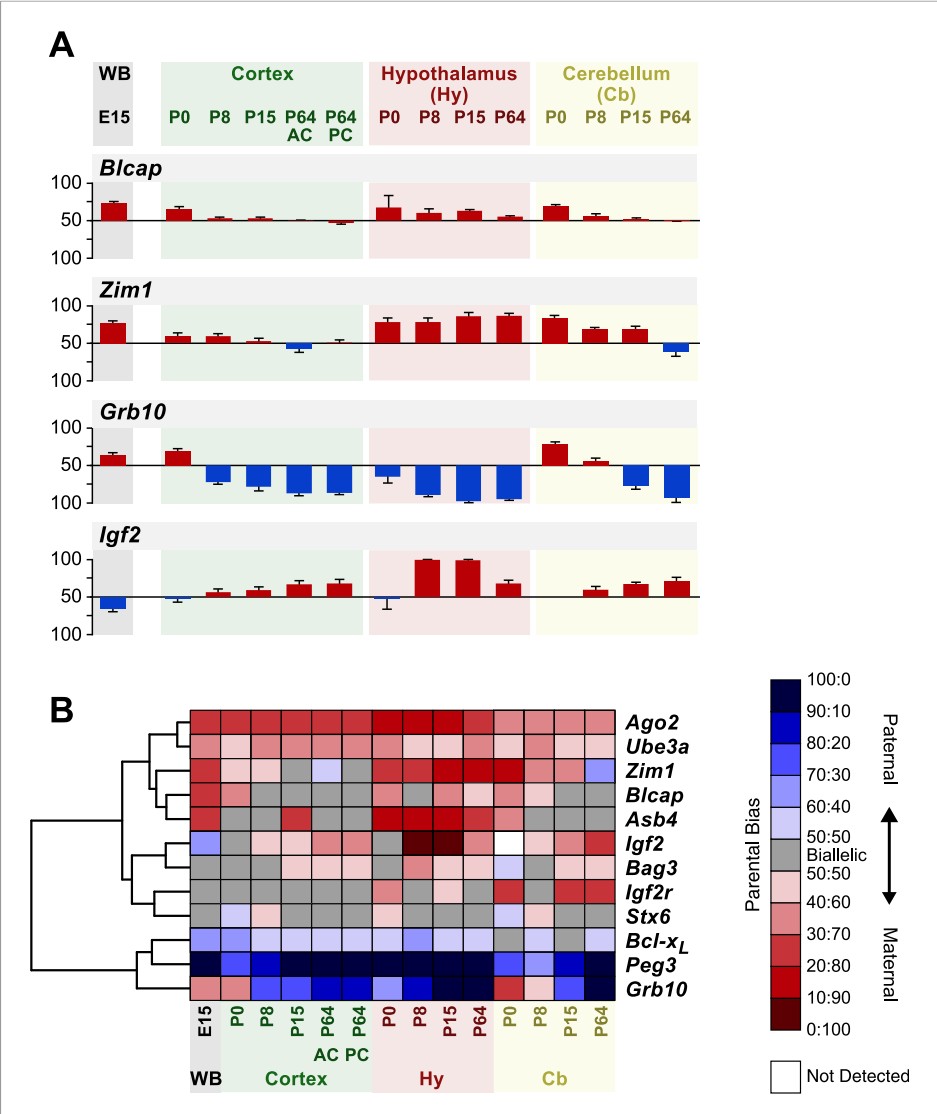

**Figure 7**. Spatiotemporal regulation of imprinted genes. (**A**) Analysis of parental biases at different time points during development of the whole brain (WB) at embryonic day 15, and of the cortex, hypothalamus, and cerebellum at four post-natal stages. Origins of bar graphs (N = 6 in each bar) represent biallelic expression. Positive values represent preferential maternal expression (colored red), and negative values represent preferential expression of the paternal allele (colored blue). (**B**) Hierarchical clustering of the heat map representing deviations from biallelic expression in imprinted genes (N = 6 in each square).

## Functional insights into the role of genomic imprinting in the apoptotic pathway

In order to gain initial insights into the functional implications of the observed patterns of parental bias across the brain, we investigated whether specific biological pathways are enriched among cerebellum-imprinted genes. The category of programmed cell death (apoptosis) includes eight genes previously shown to exhibit parent-of-origin monoallelic expression, and five genes that we have newly identified as exhibiting parentally biased expression (*Figure 8A*). Our data thus uncover the programmed cell death pathway as a prominent target of imprinting. Noticeably, most of the maternally-biased genes (*Cdkn1c*, *Kcnk9*, *Blcap*, and *Bmf*) promote apoptosis, while most of the paternally-biased genes (*Ndn*, *Peg3*, *Peg10*, *Plagl1*, and *Magel2*) inhibit apoptosis. One gene of particular interest is the paternally-biased *Bcl-x* (*Gregg et al., 2010*), which can produce two distinct

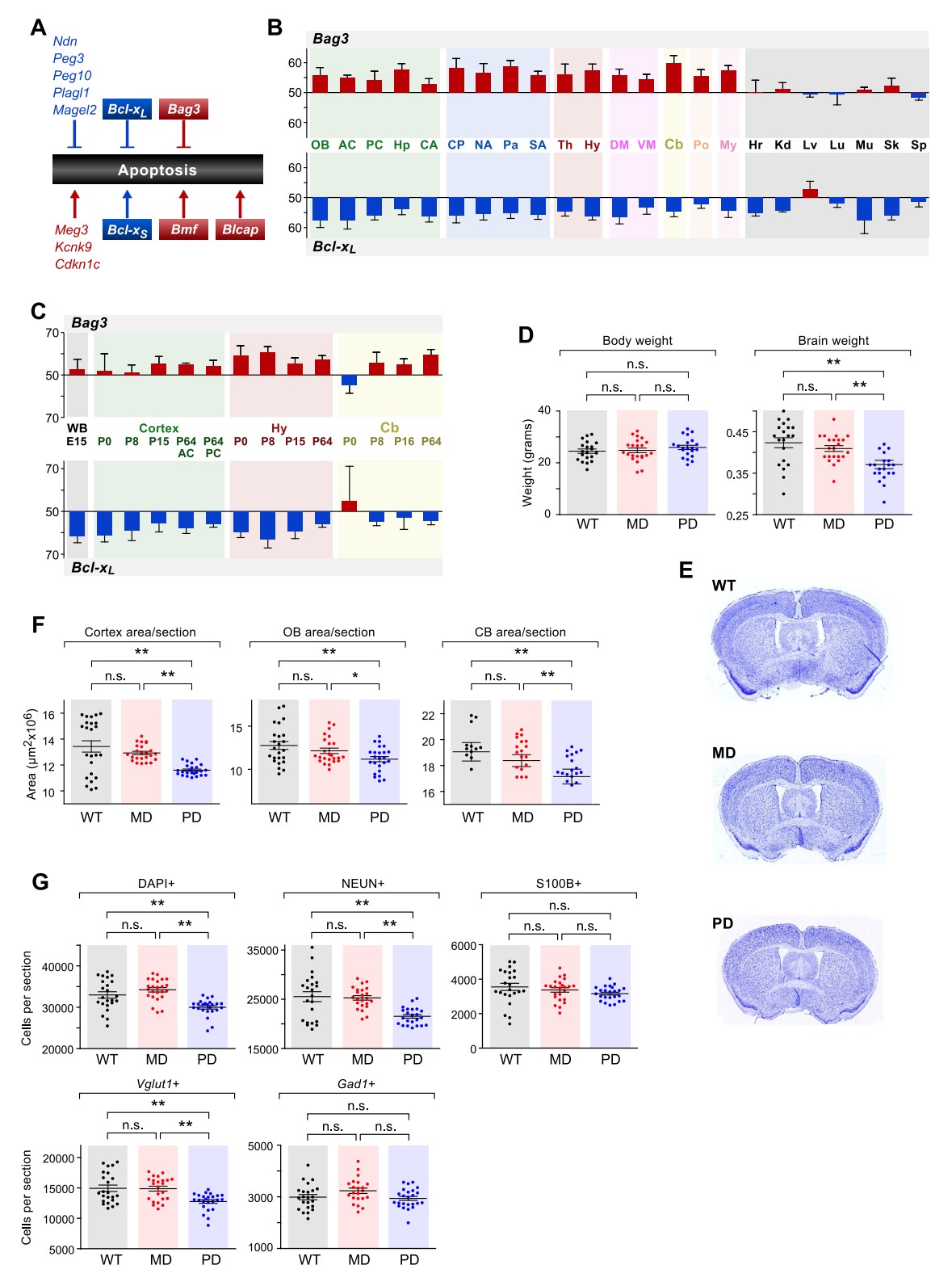

**Figure 8**. The role of genomic imprinting in the apoptotic pathway in the brain. (**A**) Imprinted genes (color coded by parental bias: red for maternal and blue for paternal) involved in the regulation of apoptosis. Colored boxes denote genes exhibiting weak to moderate parental biases. Lines with arrowheads indicate pro-apoptotic function and notched lines indicate anti-apoptotic function. (**B**) Anti-correlated spatial regulation of $Bcl-x_L$ and $Bag3$ biases in the adult brain and body (Pearson correlation coefficient = −0.56; p-value = 0.04). (**C**) Regulation of $Bcl-x_L$ and $Bag3$ biases in the developing

*Figure 8. Continued*

brain. (**D**) Body and brain weights of adult (P80) mice bearing nervous system-specific deletions of the maternal (MD), paternal (PD), or neither (wild type (WT)) allele of *Bcl-x*. Each data point represents an individual mouse. Data include both males and females, as no significant gender-specific differences were observed. (**E**) Representative images of NISSL-stained coronal sections that were used for measuring cortical areas. Tissue sections shown are from P80 male mice of each genotype. (**F**) Cross-sectional areas of cortex, olfactory bulb (OB), and cerebellum (Cb) for each genotype. Each data point represents a coronal section from a total of six P80 male mice. (**G**) Quantification within the cortex of the number of cells per section labeled with specific markers: DAPI (all cells), NEUN (neurons), S100ß (subset of glia), *Vglut1* (*Slc17a7*, subset of excitatory neurons), and *Gad1* (subset of inhibitory neurons). Each data point represents a coronal section from a total of six P80 male mice.

The following figure supplements are available for figure 8:

**Figure supplement 1**. Cell-type specific effects in the cortex of brain-specific deletion of the paternal and maternal *Bcl-x* alleles.

**Figure supplement 2**. Effects of brain-specific deletion of the paternal and maternal Bcl-x alleles.

---

protein isoforms: the anti-apoptotic *Bcl-x$_L$* and the pro-apoptotic *Bcl-x$_S$*, translated from two distinct mRNA transcripts (***Boise et al., 1993***). *Bcl-x$_L$* is by far the predominant *Bcl-x* transcript in the brain (***Krajewska et al., 2002***), where it exhibits widespread paternally-biased expression (***Figure 8B***). Also of interest is the gene *Bag3*, which was newly identified by our study as maternally-biased in the brain and biallelic in peripheral tissues (***Figure 8B***). The protein products of genes *Bcl-x$_L$* and *Bag3* have multiple functions, including some unrelated to apoptosis (***Roth and D'Sa, 2001***; ***Rosati et al., 2011***). However, a suggested function of these proteins, mediated by their mutual interaction, is to prevent the mitochondrial release of factors controlling activation of caspases and thus the irreversible commitment to undergoing apoptosis (***Jacobs and Marnett, 2009***). Our RNA-seq analysis of the cerebellum revealed that the maternal bias of *Bag3* substantially increases from P8 to P60 whereas *Bcl-x$_L$* shows a substantial decrease in paternal bias between these two time points (***Figure 8—figure supplement 1A***). Further quantification of parental biases across 16 regions of the adult brain (***Figure 8B***) shows that, with the exception of the ventral midbrain and pons, the parental biases of *Bag3* and *Bcl-x$_L$* are significantly anti-correlated (Pearson correlation coefficient = −0.56; p-value = 0.04, ***Figure 8B***). This pattern is most evident in the cortex and cerebellum, as well as in central striatal regions, caudate putamen, and hippocampus. Variation in the parental biases of *Bcl-x$_L$* and *Bag3* are also observed in our developmental analysis but an anti-correlation between the two is not as apparent (***Figure 8C***).

We next sought to investigate the functional significance of the parentally biased expression of genes within the programmed cell death pathway by analyzing the *Bcl-x* (*Bcl2l1*) gene, which has a well-characterized role in inhibiting neuronal apoptosis during brain development due to production of the *Bcl-x$_L$* protein isoform (***Roth and D'Sa, 2001***). Complete knockout of *Bcl-x* results in massive embryonic apoptosis and lethality, whereas loss of one allele has been reported to reduce brain size by ~15% (***Kasai et al., 2003***). We hypothesized that if the paternal and maternal alleles unequally contribute to the expression of *Bcl-x$_L$* in the brain, and if this differential parental contribution is of functional significance, the differential loss of each parental allele should result in distinct functional outcomes. In particular, loss of the paternal (more highly expressed) allele should result in a significantly more severe phenotype than the loss of the maternal (less expressed) allele.

To test this hypothesis, we generated deletions of the maternal and paternal alleles of *Bcl-x* specifically in the brain by transmitting a floxed *Bcl-x* allele from either the mother or the father to offspring along with a *Nestin::Cre* transgene, which is expressed specifically in the brain. The floxed *Bcl-x* allele enables CRE-dependent deletion of both the anti-apoptotic *Bcl-x$_L$* and pro-apoptotic *Bcl-x$_S$* isoforms. However, since *Bcl-x$_L$* is substantially more highly expressed in the brain (***Krajewska et al., 2002***), the brain-specific deletion of *Bcl-x* is expected to affect the *Bcl-x$_L$* isoform nearly exclusively. As expected, neither the brain-specific maternal deletion nor paternal deletion of *Bcl-x* had a significant effect on the body weight of adult mice (week 11–12, ~P80). However, we found that mice with a paternal *Bcl-x* deletion had a ~15% reduced brain weight compared to wild type (WT) littermate controls, which carried the *Nestin::Cre* transgene and two WT copies of *Bcl-x* (***Figure 8D***). In contrast, mice with a maternal *Bcl-x* deletion exhibited no significant difference in brain weight compared to WT littermates. We next asked whether the paternal *Bcl-x* deletion has observable

effects on brain morphology by measuring the cross-sectional area of three different brain structures: cortex, olfactory bulb, and cerebellum. In all three structures, significant differences in area were observed between mice with a paternal *Bcl-x* deletion and both mice with a maternal deletion and WT littermates, while no significant differences were found between mice with a maternal *Bcl-x* deletion and WT littermates (*Figure 8E,F*, and *Figure 8—figure supplement 2A,B*). Taken together, these results support the hypothesis that the paternal allele of *Bcl-x* makes a more substantial contribution to brain development than the maternal allele.

We next assessed the cell specificity of the reduced brain weight phenotype observed upon the paternal deletion of *Bcl-x*. For these analyses, we used histochemical methods to quantify total cortical cells (DAPI) as well as subsets of cells that express specific cellular markers: NEUN (RBFOX3, neurons), S100ß (subset of glia), *Vglut1* (*Slc17a7*, subset of excitatory neurons), and *Gad1* (subset of inhibitory neurons). Our data show that DAPI+, NEUN+, and *Vglut1*+ cells were significantly reduced in paternal *Bcl-x* deletion cortices compared to their maternal deletion and WT counterparts, while S100ß+ and *Gad1*+ cells were not significantly affected (*Figure 8G,H*, and *Figure 8—figure supplement 1B–D*). Cell densities between the three genotypes differed only modestly for all cell types examined (*Figure 8—figure supplement 1B*). Similar analyses of the effects of paternal and maternal *Bcl-x* deletion on lobule 6 of the cerebellum revealed significantly fewer cells in the molecular layer of paternal deletion compared to WT mice (*Figure 8—figure supplement 2C–E*). Within the molecular and Purkinje layers, *Gad1*+ neurons were significantly reduced in number in mice with a paternal *Bcl-x* deletion compared to WT littermates (*Figure 8—figure supplement 2F,G*). Together, these results reveal that deletion of the paternal allele of *Bcl-x* causes dramatic reductions in brain size and cell number that are not observed upon deletion of the maternal allele. Moreover, these effects differ according to cell type and brain region, with specific cell types affected in a given region. They therefore suggest a largely unexpected role of genomic imprinting in the regulation of the distribution of specific cell types in distinct areas of the mouse brain.

## Discussion

### Transcriptome-wide analysis of genomic imprinting in the cerebellum

In this study we used the high resolution afforded by RNA-seq to profile genomic imprinting in the mouse cerebellum with high statistical power (48 biological samples). We analyzed expression of the parental alleles at the individual transcript level, and simultaneously estimated the effects of all factors in the experimental design (age, sex, and mouse cross) on parental allelic expression. This methodological approach allows precise quantification of the entire spectrum of parental biases observed in imprinted genes, and accurately reveals the contribution of each parental allele to the overall gene expression. In addition to 74 imprinted genes that had been previously validated in earlier studies (*Morison et al., 2005*; *Tunster et al., 2013*), our analysis detects 41 novel imprinted genes that we successfully confirmed by pyrosequencing. Taken together, our approach has a precision of ~93% and our data increase the total number of reported imprinted genes in the mouse by approximately 30%.

Our approach demonstrates reliable inference of genomic imprinting when based on a large number of biological samples and a statistical method such as BRAIM that explicitly estimates the biological variability. Previous surveys of imprinting in the mouse either had only a single pair of F1i and F1r sample (*Gregg et al., 2010*) or had multiple such samples yet combined counts of reads across samples of the same cross (*Babak et al., 2008*; *Wang et al., 2008*; *DeVeale et al., 2012*) thereby overlooking biological variability and risking reaching erroneous conclusions (Simpson's paradox). A direct side-by-side comparison between our results and previously published data is not a fair comparison as our method naturally penalizes for the lower number of samples and possibly the weaker signal due to shallower sequencing. In addition, since none of the previously published high throughput screens for imprinting aimed to systematically validate all identified candidate imprinted genes, it is impossible to objectively estimate their associated false positive rates as achieved in this work.

Importantly, most of the newly identified imprinted genes, as well as a substantial number of those previously known, display weak-to-moderate parental biases rather than the more conventionally expected monoallelic expression. As the tissue samples analyzed by RNA-seq are comprised of a mixture of cell types it is impossible to distinguish whether the parental biases are present in all cell

types of the cerebellum or whether they may result from averaging monoallelic expression of given transcripts in some cell types and biallelic representation in others. Interestingly, cell-type-specific imprinting has been reported for *Gnas* in the kidney within different segments of nephrons (*Weinstein et al., 2000*), and for *Ube3a* and *Snx14* in neurons but not glial cells (*Yamasaki et al., 2003*; *Huang et al., 2014*). Clarifying the origin of parental biases is clearly warranted and will require allelic resolution in defined cell types.

Is parentally-biased expression a form of genomic imprinting? Imprinted genes are often described and even defined as monoallelically expressed in a parent-of-origin specific manner. Such definition, however, is incomplete since a substantial number of known imprinted genes appear preferentially expressed from one of the parental alleles. Indeed, the concurrent detection of both mononallelically-expressed and parentally-biased imprinted genes has been highlighted in multiple genome-wide profiles of genomic imprinting (*Babak et al., 2008*; *Wang et al., 2008*; *Gregg et al., 2010*; *DeVeale et al., 2012*; *Zou et al., 2014*). Whether parentally-biased genes correspond to the same biological entity as other imprinted genes has been discussed (*Khatib, 2007*; *Gregg, 2014*), but the regulation and functional significance of this phenomenon remains largely obscure.

Our data suggest that monoallelic and parentally biased representations share similar characteristics, and may thus represent different ranges of a common mode of differential regulation of parental allelic expression. First, the majority of imprinted genes with weak-to-moderate parental bias are located nearby or at known imprinted clusters, and we show that this localization is evolutionarily conserved, suggesting it is under purifying selection. Second, genes like *Actn1* and *Nhlrc1*, which exhibit weak parental biases and are isolated in the genome, have been reported to have differentially methylated regions (DMRs) between the parental alleles (*Calaway and Domínguez, 2012*; *Xie et al., 2012*). These DMRs may thus serve as imprinting control regions as has been shown for monoallelically expressed genes. Third, a substantial proportion of genes with weak-to-moderate biases, as well as strongly imprinted genes, show tissue and developmental-stage specificities in their parental biases. For instance, *Copg2* shows a weak maternal bias in the cerebellum (∼60% maternal expression to 40% paternal expression) while its maternal bias in the cortex is close to monoallelic (90% maternal expression to 10% paternal expression). It is therefore possible that many genes with weak-to-moderate parental bias in the cerebellum are actually strongly imprinted in other tissues or developmental stages. Finally, many of the genes with weak-to-moderate biases are implicated in the same biological processes as genes with strong biases, including cell survival and apoptosis.

## Dynamic temporal and spatial regulation of imprinted genes in the brain

Altogether, the parental bias and/or overall expression of more than half of the genes imprinted in the cerebellum are regulated according to age, with approximately 25% of cerebellum imprinted genes exhibiting changes in the magnitude of their parental bias between P8 and adult stages. Postnatal day 8 is an important milestone of cerebellar development that includes a peak of granule cell precursor proliferation, the active migration of granule cells to the inner granule layer, and the establishment of connectivity between mossy fibers, granule cells, and Purkinje cells. Interestingly, most of the regulated genes exhibit stronger parental biases and/or higher overall expression during this time point than in the mature cerebellum. This is the case for *Cdkn1c*, *Plagl1*, and *Ago2*, which are known to regulate cell survival and differentiation during embryonic development (*Cheloufi et al., 2010*; *Tury et al., 2012*; *Schmidt-Edelkraut et al., 2014*). A smaller subset of genes, including *Stx6*, *Adam23*, and *Pcdhb12*, have been shown to regulate cell motility and neuronal connectivity, suggesting that genomic imprinting is also involved in these developmental processes (*Owuor et al., 2009*; *Tiwari et al., 2011*; *Chen and Maniatis, 2013*).

Previous studies have shown differential regulation in the overall expression and, in some cases, the imprinting status of imprinted genes across body tissues and brain regions (*Albrecht et al., 1997*; *Gregg et al., 2010*; *Prickett and Oakey, 2012*). Our study adds substantial new insights into this phenomenon by investigating the regulation of parental bias of 28 known and novel imprinted genes, across 16 brain regions and 7 somatic non-neural tissues. Three interesting observations emerge from this analysis. First and most noticeably, many genes tested appear uniquely imprinted in the brain. Second, a subset of the genes tested show significant variability in the degree of parental bias across different brain regions. Third, genes within an imprinted cluster, as well as occasional genes located on distinct chromosomes, display interesting similarities in the patterns of paternal bias across brain

regions. Additionally, for a subset of imprinted genes, we observed regulation of parental bias according to both the developmental stage and the brain region, suggesting that differences in parental bias observed throughout the adult brain may originate during the lineage specification of distinct brain regions, and may also result from different cellular compositions of the tissues analyzed.

The substantial spatial and temporal regulation of parent-of-origin allelic expression observed in our study, particularly between brain and non-brain tissues, raises the question of the specific mechanisms that may control these dynamics. For most imprinted genes, monoallelic or parentally biased expression has been shown to be primarily dependent on DNA methylation marks established in one of the parental germ lines, and maintained post-fertilization in the parental alleles of the embryo and mature organism (*Bartolomei and Ferguson-Smith, 2011*). In a smaller subset of imprinted genes, the differential methylation of the parental alleles is established in somatic tissues (*Wang et al., 2014a*). Other genes such as *Dlk1* and *Cdh15* become biallelically expressed in specific cell types after gaining DNA methylation on the normally unmethylated allele (*Ferrón et al., 2012*; *Proudhon et al., 2012*). Based on these established mechanisms, it is possible that the observed spatial and temporal regulation of parental bias depends on cell-lineage-specific acquisition of DNA methylation or other epigenetic modifications in one of the parental alleles. It is also possible that changes in parental bias result from loss of methylation or perhaps hydroxymethylation (*Guo et al., 2011*) of DMRs. Alternatively, spatial and temporal variations in parental biases may depend on the presence or absence of chromatin regulators directed by DNA methylation, which in turn control necessary steps to achieve preferential expression of one of the parental alleles (*Kulinski et al., 2013*), or may rely on the use of brain- and brain region-specific promoter elements. If the parental bias of a certain gene changes according to cell-type, different cellular compositions may also contribute to the observed variability across tissues. This possibility is particularly intriguing in cases where parental biases change according to brain region.

## Relation between parental bias and gene expression

What, if any, is the functional significance of the dynamic regulation of parental biases identified in our study? We reasoned that the notion of a significant functional contribution of the observed parental bias to normal brain function and development should be supported by at least three elements: first, the parental bias of a given gene must be evolutionary conserved, second, changes in parental bias of a given gene should be consistent with changes in overall expression level, and third, uniparental deletions should lead to different phenotypes. Our data partially support these three notions. First, although we did not investigate parental biases in other species, the evolutionary conserved micro-synteny of biased genes suggests that natural selection may operate against the disruption of the biased expression. Second, an attractive hypothesis underlying parental allelic biases may lie on the requirement for different brain regions, and possibly neuronal types, to tightly regulate the expression of certain genes for proper cellular or network function. Converging evidence from mental disorders associated with slight over- or under-regulation of certain genes (*Dölen et al., 2007*; *Ramocki and Zoghbi, 2008*) suggests that normal brain function may indeed require precise gene titration. Our study reveals that certain genes undergo striking variations in parental bias from one brain region to the next, or from one developmental stage to the other. For example, *Ago2* shows a modest 60% maternal bias in the cerebellum and olfactory bulb, but a robust ~80% maternal expression in the cortex. Similarly, *Zim1* shows biallelic expression in cortical regions, 85% maternal bias in the pallidum and hypothalamus, and 60% paternal expression in the cerebellum. These data, however, were collected using pyrosequencing, which does not inform on the overall gene expression level, hence such spatial changes in parental biases may, or may not, be associated with up- or down regulation of genes.

The RNA-seq data enabled us to detect a positive correlation between developmental changes in the magnitude of parental biases and overall gene expression for most imprinted genes. Moreover, we find evidence that in the majority of cases, an age-regulated increase in parental bias (age effect PP > 0.95) is accompanied by a significant up-regulation in the expression level of the preferred allele and thus, an increase in parental bias correlates with an increase in overall expression level. This indicates that changes in parental bias and overall expression are coordinated according to development stage and brain region, which may effectively modulate gene dosages in individual cells. In addition, or alternatively, changes in parental biases may arise from different cellular distributions across developmental and adult brain areas. Both scenarios

commonly point to a highly dynamic regulation of genomic imprinting according to neuronal types. Finally, the distinct phenotypes observed in mice bearing paternal and maternal deletions of *Bcl-x* argue strongly that, even for genes displaying weak-to-moderate biases, the two parental alleles may differ in their functional significance.

## The functional significance of *Bcl-x$_L$* paternal bias

Several imprinted genes are known to play important roles in the apoptotic pathway, and we uncovered additional imprinted genes with apoptosis-related functions. Apoptosis plays an important role during brain development where it regulates cell population size and refines neuronal circuits by removing poorly connected or non-functional cells (*Buss et al., 2006*). Accordingly, most of the imprinted apoptosis-related genes observed in our study exhibit stronger overall expression and parental biases in the developing cerebellum than in the adult. We also observe a significant number of pro- and anti-apoptotic genes that are differentially expressed according to brain region. Considering that most neurons in the adult brain are post-mitotic, the functional significance of such spatial regulation is still unclear. Some imprinted genes, however, are known to be highly pleiotropic (e.g., *Cdkn1c*, *Tury et al., 2012*), hence alternative functions of these imprinted apoptotic genes in the adult brain cannot be ruled out.

The *Bcl-x* gene, which performs a critical anti-apoptotic function in developing neurons (*Kuan et al., 2000*), exhibits a moderate bias in most adult brain regions and a slightly larger bias in the developing brain. Remarkably, the deletion of the paternal allele of *Bcl-x* results in a significant reduction in brain size and brain cell number compared to littermate controls, while deletion of the maternal allele has no significant effect. In addition, our histochemical analyses of brain regions from mutant mice and littermate controls reveal intriguing evidence for distinct functional consequences of the paternal *Bcl-x* deletion on specific cell types within a given region of the brain. In the cortex, we find that the paternal *Bcl-x* deletion preferentially affects neurons, but not glia, and more specifically, *Vglut1*-positive excitatory, but not *Gad1*-positive inhibitory neurons. In contrast, the *Gad1*-positive Purkinje cells are clearly affected by the paternal deletion in the cerebellum.

An unambiguous interpretation of these results will require future investigations to identify the origin of the paternal bias of *Bcl-x$_L$*. If the observed parental bias of *Bcl-x$_L$* within a given brain region is shared within all cells of the tissue, the deletion of the paternal allele should result in a larger reduction of *Bcl-x$_L$* expression than deletion of the maternal allele, and the observed phenotype may reveal a high cellular sensitivity to the dosage of *Bcl-x$_L$* at the cellular level. In contrast, if the observed parental bias of *Bcl-x$_L$* in a given brain area originates from a mixture of cells expressing exclusively the paternal allele of *Bcl-x$_L$* with cells expressing both parental alleles equally, the deletion of the paternal allele should result in the complete absence of *Bcl-x$_L$* in monoallelically-expressing cells, likely resulting in cell death. Deletion of either parental allele of *Bcl-x$_L$* in biallelically-expressing cells may only generate a mild phenotype if any.

At the level of the organism, our data uncover how paternal and maternal alleles of *Bcl-x* make vastly different contributions to brain development, a result that has profound implications for the analysis of parentally-inherited polymorphisms in human health.

## Materials and methods

### Animal subjects

F1 hybrids were generated by reciprocally crossing C57Bl/6J and Cast/EiJ mouse strains, where we denote by F1i an F1 hybrid derived from a C57Bl/6J father and a Cast/EiJ mother, and by F1r an F1 hybrid derived from a Cast/EiJ father and a C57Bl/6J mother. For the RNA-seq data, we used 48 animal subjects covering both crosses, both sexes, and two age groups: developing animals sacrificed at postnatal day 8, denoted as P8, and adult animals sacrificed in the range of postnatal days 56–64, denoted as P60. Our experimental design is balanced thus having three factors: cross, sex, and age, with six animal replicates in each factor block.

### Reference genome and transcriptome data

We created a C57Bl/6J, Cast/EiJ diploid genome by incorporating C57Bl/6J and Cast/EiJ single nucleotide polymorphisms and indels (obtained from the Mouse Genome Project:

ftp://ftp-mouse.sanger.ac.uk/REL-1303-SNPs_Indels-GRCm38) into the *M. musculus* GRCm38 reference genome sequence. We additionally created a transcriptome annotation set as follows. We first downloaded the Gencode (*Engström et al., 2013*; *Steijger et al., 2013*) M2 mouse main gene annotation (gencode.vM2.annotation) general transfer format (GTF) file and removed from it the following RNA types: Mt_rRNA, Mt_tRNA, miRNA, rRNA, snRNA, snoRNA, Mt_tRNA_pseudogene, tRNA_pseudogene, snoRNA_pseudogene, snRNA_pseudogene, scRNA_pseudogene, rRNA_pseudogene, miRNA_pseudogene, as they are not supposed to be enriched in our RNA libraries. In order to include as an extensive set of transcripts as possible and to specifically cover retroposed genes due to their known involvement in genomic imprinting (*McCole and Oakey, 2008*), we followed these steps to add additional annotated transcripts to gencode.vM2. annotation. First, we downloaded the Gencode Retrotransposed (gencode.vM2.2wayconspseudos) GTF file and the ucscRetroInfo2 mm10 mouse genome assembly retrogenes annotation file from the University of California Santa Cruz (UCSC) genome browser (*Karolchik et al., 2014*). We eliminated any redundancy between the two transcript sets by selecting the longest transcript between any two transcripts represented in both files. Following that, we eliminated any redundancy between that retrogene set and gencode.vM2.annotation transcript set by selecting the longest transcript between any two transcripts represented in both sets. In order to remove any redundancy between our retrogene set and single-exon protein coding transcripts (which is a structural prominent feature of retrogenes) in gencode.vM2.annotation we kept the longest of any intersecting protein-coding single-exon transcript in gencode.vM2.annotation and retrogene in our retrogenes set. Subsequently, we added all transcripts from the UCSC knownGene mm10 mouse genome assembly annotation file, which are not indicated to be represented in the gencode.vM2.annotation set. Finally, we added all functional RNAs from the functional RNA database (fRNAdb, *Mituyama et al., 2009*), which did not intersect with any of the transcripts in the set we generated in the previous steps and is longer than our 59 bp read length. Altogether, this amounted in 148,120 transcripts from 92,965 genes, comprised of 60,978 protein coding transcripts, 87,142 non-coding transcripts, among which 16,404 are pseudogenes, 15,538 are retrogenes, 2518 are lincRNAs, and the remaining 52,682 are of other miscellaneous types. We then used the UCSC liftOver utility to generate a C57Bl/6J, Cast/EiJ diploid transcriptome set from our generated transcriptome set. We note that many retrogenes are highly redundant in sequence with their paralogs. In the case of short, single-end read RNA-seq data, the accuracy of their expression level estimates would therefore, by and large, be low. For this reason, retrogenes added from the ucscRetroInfo2 retrogenes annotation, which are of lower certainty than retrogenes in the Gencode annotation, are indicated in *Supplementary file 1A–F*.

## RNA-seq sample preparation

RNA was isolated from tissues of interest using the Trizol reagent (Life Technologies, Carlsbad, CA) according to the manufacturer instructions and further purified using DNase I digestion and the RNeasy kit (Qiagen, Netherlands). We required samples to have RNA integrity score of above 9, according to the Agilent (Santa Clara, CA) 2100 Bioanalyzer, to be used for RNA-seq library preparation. For each sample we used 3 µg of total RNA to prepare libraries according to the Illumina (San Diego, CA) Tru-Seq RNA Kit v2 sample preparation protocol. Sample purity and integrity was confirmed using the Agilent 2100 Bioanalyzer. We selected an average fragment size of 250 bp. Each animal subject was used to prepare a single library and was sequenced on an individual lane generating 59 bp single-end reads. The average read depth across our samples was 168,991,714.5.

## Processing of RNA-seq data

Each RNA-seq library was first subjected to quality and adapter trimming using the Trim Galore utility (http://www.bioinformatics.babraham.ac.uk/projects/trim_galore) with stringency level 3. We then mapped each of our C57Bl/6J×Cast/EiJ hybrid RNA-seq sequenced libraries to the C57Bl/6J, Cast/EiJ diploid genome and transcriptome splice junctions using STAR RNA-seq aligner (*Dobin et al., 2013*) allowing a maximum of three mismatches. Specifically, we mapped the data twice where after the first mapping step we incorporated valid splice junctions which were reported by STAR to exist in our RNA-seq data. We note that restricting the number of allowed mismatches to one had no apparent effect on the remaining downstream analyses. Subsequent to the second STAR mapping step we

transformed our genomic alignments to transcriptomic alignments and thus filtered any alignments that did not map to our transcript set using custom code. We note that by doing so we allow the reads to unbiasedly align to their best locations in the splice-junction aware genome and subsequently keep our alignments of interest as opposed to aligning the read data directly to the transcriptome (e.g., *Roberts and Pachter, 2013*). Following that, we estimated the expression levels with their respective uncertainties of each transcript in our C57Bl/6J, Cast/EiJ diploid transcriptome using MMSEQ (*Turro et al., 2011*). MMSEQ uses a Bayesian model for estimating expression levels and therefore computes a posterior distribution of the expression levels of each transcript in fragment per kilobase per million (FPKM) units. We first transformed these posterior FPKM samples to TPM units as TPM units were shown to be less biased and more interpretable (*Wagner et al., 2012*). We set the minimum expression TPM cutoff to 0.01. While this is a very low expression level, it still enables to detect genes with parentally biased expression, albeit with lower accuracy, in the estimated fraction of the preferentially expressed allele (see 'Results'). In any RNA-seq sample, any transcript for which its MMSEQ posterior median TPM was lower than 0.01 was set to 0.01 (which we thus used as the minimal measureable expression level and therefore avoid taking the logarithm of 0). At a 0.01 TPM expression level cutoff we detect and validate genes with parentally biased expression, although at the very low expression levels the parental biases estimated by RNA-seq strongly deviate from those estimated by pyrosequencing and there is a higher fraction of false positives (*Figure 1—figure supplement 1C*). This indicates that the accuracy of parental biases estimated by RNA-seq is limited at this range of low expression levels. Accordingly, the distribution of parental biases at very low expression levels does not follow the bimodal shape of parental biases of genes expressed at higher levels (*Figure 1—figure supplement 1B*). We therefore empirically defined a more stringent expression level cutoff corresponding to where the discrepancy between the parental biases estimated by RNA-seq and pyrosequencing drops dramatically (dashed line in *Figure 1—figure supplements 1C* and correspondingly in *Figure 1—figure supplements 1B*).

Using an extensive transcriptome annotation set has the advantage of estimating the expression levels (and therefore testing for parentally-biased expression) of as many known transcripts that are expressed in the tissue from which RNA was purified, as possible. However, it is very likely that highly similar transcripts (e.g., NAGNAG alternative splice forms, *Bradley et al., 2012*) will not be distinguishable by the read data. This, in turn, would increase the expression level uncertainties of such lowly identifiable transcripts and would therefore reduce power when testing whether their expression is parentally biased. Ideally, one would detect such lowly identifiable transcripts and combine their expression level estimates into that of a single merged transcript, indicating that either or all of them are expressed. Such combined transcripts would therefore have lower expression level uncertainty than their constituents, which would therefore increase power when testing whether their expression is parentally biased. To achieve this, we adopted the approach of *Turro et al. (2014)* for combining lowly identifiable transcripts based on the posterior correlation of their expression level estimates, tailored for a diploid transcriptome case. In this approach, for any given RNA-seq sample we compute the Pearson correlation coefficient of the posterior TPM samples of any pair of transcripts from the same locus and the same allele. Subsequently, if the mean Pearson correlation coefficient across all RNA-seq samples for a pair of transcripts in both alleles is lower than a defined cutoff (which we empirically set to −0.25), we combine each of these pairs into a single combined transcript. This process continues iteratively until no pair of transcripts (or pairs of already combined transcripts) can be further combined. This consistency between the alleles in the combining process ensures that the resulting combined transcripts are identical for the two alleles and can therefore be tested for parentally biased expression.

## BRAIM

For inferring whether a given transcript is imprinted, we define our estimand of interest as the difference in the expression level between its paternal and maternal alleles, that is, the parental bias. Intuitively, if the parental bias is approximately zero across all samples we would conclude that the transcript is not imprinted. The reality of RNA-seq data, however, is more complicated than that. First, as mentioned above, we do not obtain accurate estimates of expression levels but rather estimates with uncertainty. Next, our experimental design may include inherent factors that can affect allele-specific expression levels to various extents, such as the mouse cross, sex, and age. Whether of

interest or not, the effects of these factors need to be explicitly accounted for. In addition, even though our nearly genetically identical animal subjects are treated with similar conditions, thereby minimizing effects of any additional factors to the ones specified above (such as environment), we still expect biological variability in RNA levels across our subjects (e.g., due to litter effects). Finally, we would expect technical variability across experiments to add to the biological variability, yet unless addressed explicitly (e.g., sequencing the same RNA library as technical replicates), the two are indistinguishable.

To address all of these issues we developed a statistical model for inferring genomic imprinting from our experimental design for every transcript in our annotation set. Specifically, we have chosen to extend the Bayesian variable selection regression model of *Chipman et al. (1997)* by accounting for the measurement error in the response, as uncertainties are naturally propagated in a Bayesian framework. In our model, the response of sample $j$ ($\widehat{y}_j$, where $j = 1,..., n$ samples) for a certain transcript is the mean posterior difference between the paternal and maternal natural log posterior TPM samples, that is,

$$\widehat{y}_j = \frac{\sum_{s=1}^{S}\log\left(TPM_j^{(s)}(paternal) + c\right) - \log\left(TPM_j^{(s)}(maternal) + c\right)}{S},\tag{1}$$

where $TPM_j^{(s)}(paternal)$ and $TPM_j^{(s)}(maternal)$ denote the paternal and maternal posterior sample $s$ of $S$ posterior samples, respectively for RNA-seq sample $j$. By $c$ we denote the minimum detectable expression cutoff of that RNA-seq sample (described above) that we add to the TPM in order to avoid taking the log of zero. Since regression parameters are sensitive to the scale of inputs (*Gelman, 2008*) yet we wish to have a common interpretation for all transcripts we fit our model to, we divide the response $\widehat{y}_j$ by the standard deviation of $\widehat{y}_j$'s across all $j \in n$ samples and denote this scaled response as $\widehat{y}_j'$.

We define the uncertainty (or measurement error) of the response as:

$$\widehat{\varepsilon}_j = \frac{\sum_{s=1}^{S}\left[\log\left(TPM_j^{(s)}(paternal) + c\right) - \log\left(TPM_j^{(s)}(maternal) + c\right) - \widehat{y}_j\right]^2}{S-1}.\tag{2}$$

This therefore represents the posterior variance of the estimated parental bias for RNA-seq sample $j$.

We thus model:

$$\left(\widehat{y}_1', ..., \widehat{y}_j', ..., \widehat{y}_n'\right) = y'|z \sim MVN(z, E).\tag{3}$$

We denote $z = (z_1,..., z_j,..., z_n)$ as the unobserved (latent) true value of the response, and $E = diag(\widehat{\varepsilon})$ as the covariance matrix of the response errors $\widehat{\varepsilon} = (\widehat{\varepsilon}_1, ..., \widehat{\varepsilon}_j, ..., \widehat{\varepsilon}_n)$. We denote by $n$ the total number of samples. In our experimental design, each of our $k$ factors, namely, cross, sex, and age has two levels, and each factor is represented by $n_{rep}$ replicates, therefore $n$ corresponds to $2^k \times n_{rep}$ observations. We model the, unobserved, true value of the response, $z$, as:

$$z|\beta, \sigma^2 \sim MVN(X\beta, \Sigma).\tag{4}$$

We denote the experimental design matrix by $X = (X_1,..., X_p)$, the regression factor parameters by $\beta = (\beta_1,..., \beta_p)$, and across-samples errors as $E = diag(\sigma^2)$. In our model we define the parental bias as the mean response across all samples, which is the intercept of the regression. Therefore, $X_1$ is a column vector of 1's, $\beta_1$ is the parameter that quantifies the effect of the parental bias, and $p = k + 1$ is the number of factors. Since we are interested in testing whether each effect $i$ is significant (i.e., whether $\beta_i$ is significantly different from zero), we model the $\beta$'s as a mixture of two normals, such that the first normal is centered around zero with a small variance, representing a non-significant effect, and the second normal is centered around the estimated $\beta_i$ and considers the effect as significant:

$$\beta_i|\tau_i, c_i, \delta_i = \begin{cases} N\left(0, \tau_i^2\right), & \delta_i = 0 \\ N\left(0, (c_i\tau_i)^2\right), & \delta_i = 1 \end{cases}.\tag{5}$$

$\delta_i$ can be interpreted as a random variable that indicates whether factor $i$ has an effect on the response. $\delta$'s are therefore modeled as Bernoulli i.i.d., with probability $p_i$ for each $\delta_i$:

$$\pi(\delta) \propto \prod_{i \in p} p_i. \tag{6}$$

The $\tau$'s and $c$'s are hyper-parameters for scaling the two normals and $p_i$ is the prior probability that factor $i$ has a significant effect on the response. Finally, we put a conjugate prior on $\sigma^2$:

$$\sigma^2 \sim IG(\nu/2, \nu\lambda/2), \tag{7}$$

where $\nu$ and $\lambda$ are hyper-parameters for the location and scale of $\sigma^2$, respectively.

## Bayesian inference and Gibbs sampling

In order to sample each of the parameters we employ a Gibbs sampling strategy where we seek to sample from the joint full posterior distribution. The joint posterior distribution of the observed data, $y'$, the covariance matrix of the response errors, $E = diag(\widehat{\varepsilon})$, and the parameters and latent variables, $\theta = (z, \beta, \sigma^2, \delta)$ is:

$$f(\widehat{y}', E; \theta) = f(\widehat{y}'|z, E)\pi(z|\beta, \sigma^2)\pi(\beta|\delta, \sigma^2)\pi(\delta)\pi(\sigma^2), \tag{8}$$

where,

$$f(\widehat{y}'|z, E) \propto |E|^{-1/2} \exp\left\{-\frac{1}{2}(\widehat{y}' - z)^T E(\widehat{y}' - z)\right\}, \tag{9}$$

$$\pi(z|\beta, \sigma^2) \propto |\Sigma|^{-1/2} \exp\left\{-\frac{1}{2}(z - X\beta)^T \Sigma^{-1}(z - X\beta)\right\}, \tag{10}$$

$$\pi(\beta|\delta, \sigma^2) \propto (\sigma)^{-P} \exp\left\{-\frac{1}{2}\beta^T \Sigma_\delta^{-1} \beta\right\}, \tag{11}$$

where,

$$\Sigma_\delta = diag(\sigma c_i^{\delta_i} \tau_i), \tag{12}$$

$$\pi(\sigma^2) \propto (\sigma^2)^{-\frac{\nu}{2}+1} \exp\left\{-\frac{\nu\lambda}{2\sigma^2}\right\}, \tag{13}$$

and,

$$\pi(\delta) \sim multinom(p). \tag{14}$$

In a Gibbs sampling strategy each parameter is iteratively sampled from its conditional posterior distribution given all other parameters. The conditional posterior distribution of $z$ is:

$$f(z|\sigma^2, \beta, \widehat{y}') \propto f(\widehat{y}'|z, E)\pi(z|\beta, \sigma^2) \propto$$

$$|E|^{-12} \exp\left\{-\frac{1}{2}(\widehat{y}' - z)^T E(\widehat{y}' - z)\right\} \times \tag{15}$$

$$|\Sigma|^{-12} \exp\left\{-\frac{1}{2}(z - X\beta)^T \Sigma^{-1}(z - X\beta)\right\},$$

where by dropping the terms not involving $z$ we get:

$$f(z|\sigma^2, \beta, \widehat{y}') \propto \exp\left\{-\frac{1}{2}\left[(\widehat{y}' - z)^T E(\widehat{y}' - z) + (z - X\beta)^T \Sigma^{-1}(z - X\beta)\right]\right\}. \tag{16}$$

We use the proposition that if:

$$f(z|\mu, y, S, \Sigma) \propto \exp\left\{-\frac{1}{2}\left[(z-\mu)^T S^{-1}(z-\mu) + (y-z)^T \Sigma^{-1}(y-z)\right]\right\}, \tag{17}$$

then (see proof in *Gelman et al., 2013*),

$$z \sim MVN(\mu_z, \Lambda_z), \tag{18}$$

where,

$$\mu_z = \Lambda_z\left(S^{-1}\mu + \Sigma^{-1}y\right), \tag{19}$$

and,

$$\Lambda_z = \left(S^{-1} + \Sigma^{-1}\right)^{-1}. \tag{20}$$

We thus obtain,

$$f\left(z|\sigma^2, \beta, \widehat{y}'\right) \propto \exp\left\{-\frac{1}{2}\left[(z-\mu_z)^T \Lambda_z^{-1}(z-\mu_z)\right]\right\}, \tag{21}$$

where,

$$\mu_z = \Lambda_z\left(\Sigma^{-1}X\beta + E^{-1}\widehat{y}'\right), \tag{22}$$

and,

$$\Lambda_z = \left(E^{-1} + \Sigma^{-1}\right)^{-1}, \tag{23}$$

and therefore,

$$z|\sigma^2, \beta, \widehat{y}' \sim MVN(\mu_z, \Lambda_z). \tag{24}$$

The conditional posterior distribution of $\beta$ is:

$$f\left(\beta|z, \sigma^2, \delta\right) \propto \pi\left(z|\beta, \sigma^2\right)\pi\left(\beta|\delta, \sigma^2\right) \propto$$
$$|\Sigma|^{-1/2} \exp\left\{-\frac{1}{2}(z-X\beta)^T \Sigma^{-1}(z-X\beta)\right\} \times \sigma^{-p} \exp\left\{-\frac{1}{2}\beta^T \Sigma_\delta^{-1}\beta\right\}, \tag{25}$$

where by dropping the terms not involving $\beta$ we get:

$$f\left(\beta|z, \sigma^2, \delta\right) \propto \exp\left\{-\frac{1}{2}\left[(z-X\beta)^T \Sigma^{-1}(z-X\beta) + \beta^T \Sigma_\delta^{-1}\beta\right]\right\}. \tag{26}$$

We use the proposition that if:

$$y|\beta, \Sigma \sim MVN(X\beta, \Sigma), \tag{27}$$

and,

$$\beta|D \sim MVN(0, D), \tag{28}$$

then (see proof in *Lindley and Smith, 1972*),

$$\beta|y, \Sigma, D \sim MVN\left(\mu_\beta, \Lambda_\beta\right), \tag{29}$$

where,

$$\mu_\beta = \Lambda_\beta X^T \Sigma^{-1} y, \tag{30}$$

and,

$$\Lambda_\beta = \left(X^T \Sigma^{-1} X + D^{-1}\right)^{-1}. \tag{31}$$

We thus obtain,

$$f\left(\beta | z, \sigma^2, \delta\right) \propto \exp\left\{-\frac{1}{2}\left[(\beta - \mu_\beta)^T \Lambda_\beta^{-1}(\beta - \mu_\beta)\right]\right\}, \tag{32}$$

where,

$$\mu_\beta = \Lambda_\beta\left(X^T \Sigma^{-1} z\right), \tag{33}$$

and,

$$\Lambda_\beta = \left(X^T \Sigma^{-1} X + \Sigma_\delta^{-1}\right)^{-1}, \tag{34}$$

and therefore,

$$\beta | z, \sigma^2, \delta \sim MVN\left(\mu_\beta, \Lambda_\beta\right). \tag{35}$$

The conditional posterior distribution of $\sigma^2$ is:

$$f\left(\sigma^2 | z, \beta, \delta\right) \propto \pi\left(z | \beta, \sigma^2\right) \pi\left(\beta | \delta, \sigma^2\right) \pi\left(\sigma^2\right) \propto$$

$$\left[\sigma^2\right]^{-\frac{n}{2}} \exp\left\{-\frac{1}{2\sigma^2}(z - X\beta)^T(z - X\beta)\right\} \times [\sigma]^{-p} \exp\left\{-\frac{1}{2\sigma^2}\beta^T\Sigma_\delta^{-1}\beta\right\} \times$$

$$\left[\sigma^2\right]^{-\frac{\nu}{2}} \exp\left\{-\frac{\nu\lambda}{2\sigma^2}\right\}, \tag{36}$$

where by dropping the terms not involving $\sigma^2$ we get:

$$f\left(\sigma^2 | z, \beta, \delta\right) \propto \left[\sigma^2\right]^{-\frac{(n+p+\nu)}{2}-1} \exp\left\{-\frac{1}{2\sigma^2}\left[\nu\lambda + (z - X\beta)^T(z - X\beta) + \beta^T\Sigma_\delta^{-1}\beta\right]\right\}. \tag{37}$$

Therefore,

$$\sigma^2 | z, \beta, \delta \sim IG\left(\frac{1}{2}(n + p + \nu), \frac{1}{2}\left[\nu\lambda + (z - X\beta)^T(z - X\beta) + \beta^T\Sigma_\delta^{-1}\beta\right]\right). \tag{39}$$

The conditional posterior distribution of $\delta$ after dropping the terms not involving $\delta$ is:

$$f\left(\delta | z, \beta, \sigma^2\right) \propto \pi\left(\beta | \delta, \sigma^2\right)\pi(\delta). \tag{40}$$

The conditional posterior distribution of $\delta$, however, is unknown, and therefore it is more suitable to sample each $\delta_i$ independently, given the set $\delta_{[-i]} = \{\delta_1,.., \delta_{i-1}, \delta_{i+1},..., \delta_p\}$. Using *Equation 40* we get:

$$f\left(\delta_i | \delta_{[-i]}, z, \beta, \sigma^2\right) \propto \pi\left(\delta_i | \delta_{[-i]}\beta, \sigma^2\right)\pi\left(\delta_i, \delta_{[-i]}\right). \tag{41}$$

Therefore,

$$p\left(\delta_i = 1 | \delta_{[-i]}, z, \beta, \sigma^2\right) =$$

$$\frac{\pi\left(\beta | \delta_i = 1, \delta_{[-i]}, \sigma^2\right)\pi\left(\delta_i, \delta_{[-i]}\right)}{\pi\left(\beta | \delta_i = 1, \delta_{[-i]}, \sigma^2\right)\pi\left(\delta_i = 1, \delta_{[-i]}\right) + \pi\left(\beta | \delta_i = 0, \delta_{[-i]}, \sigma^2\right)\pi\left(\delta_i = 0, \delta_{[-i]}\right)}$$

$$= \frac{\pi\left(\delta_i, \delta_{[-i]}\right)}{\pi\left(\delta_i = 1, \delta_{[-i]}\right) + \frac{\pi\left(\beta | \delta_i = 0, \delta_{[-i]}, \sigma^2\right)}{\pi\left(\beta | \delta_i = 1, \delta_{[-i]}, \sigma^2\right)}\pi\left(\delta_i = 0, \delta_{[-i]}\right)}, \tag{42}$$

where $\frac{\pi(\beta | \delta_i = 0, \delta_{[-i]}, \sigma^2)}{\pi(\beta | \delta_i = 1, \delta_{[-i]}, \sigma^2)}$ is the ratio of the normal mixture for $\beta$ from *Equation 11*.

Our Gibbs sampler therefore follows these steps:

1. Initialize the parameters.
2. Iterate until convergence on sampling each parameter from its conditional posterior distribution:

a. Sample $z|\sigma^2, \beta, \widehat{y}' \sim MVN(\mu_z, \Lambda_z)$ (*Equations 18, 22, 23*).
b. Sample $\beta|y, \Sigma, D \sim MVN(\mu_\beta, \Lambda_\beta)$ (*Equations 29, 33, 34*).
c. Sample $\sigma^2|z, \beta, \delta \sim IG\left(\frac{1}{2}(n+p+\nu), \frac{1}{2}\left[\nu\lambda + (z - X\beta)^T(z - X\beta) + \beta^T \Sigma_\delta^{-1}\beta\right]\right)$.
d. For $i = 1, \ldots, p$, sample $\delta_i$ according to *Equation 42*.

In all our analyses we ran our Gibbs sampler for 10,000 iterations discarding the first 1000 as burn-in.

In our experimental design all factors have binary levels: parental effect: paternal and maternal; cross effect: F1i and F1r; sex effect: male and female; and, age effect: P8 and P60. Since in this study we are interested in contrasting the two levels in each factor we set the $X$ matrix cross, sex, and age columns to 1 for F1i's, males, and P8's, and to −1 for F1r's, females, and P60's. Each of the $\beta$ parameters, which quantifies the effect of the corresponding factor, can therefore be split into the respective levels of the factor. For example, the $\beta$ which quantifies the parental effect corresponds to the mean effect across samples. Therefore:

$$\widehat{\beta} = \frac{\sum_{j=1}^n y_j}{n} = \frac{\sum_{j=1}^n \left[\log\left(TPM_j(paternal)\right) - \log\left(TPM_j(maternal)\right)\right]}{n}$$
$$= \frac{\log\left[\prod_{j=1}^n \left(\frac{TPM_j(paternal)}{TPM_j(maternal)}\right)\right]}{n}. \tag{43}$$

Therefore:

$$\exp\left(\widehat{\beta}\right) = \frac{\sqrt[n]{\prod_{j=1}^n TPM_j(paternal)}}{\sqrt[n]{\prod_{j=1}^n TPM_j(maternal)}}. \tag{44}$$

The geometric means $\sqrt[n]{\prod_{j=1}^n TPM_j(paternal)}$ and $\sqrt[n]{\prod_{j=1}^n TPM_j(maternal)}$ approximate the median paternal and maternal expression across samples. Denoting the overall expression as:

$$\sqrt[n]{\prod_{j=1}^n TPM_j(paternal)} + \sqrt[n]{\prod_{j=1}^n TPM_j(paternal)} = 1, \tag{45}$$

allows us to represent the median paternal and maternal fractions in terms of $\widehat{\beta}$:

$$\sqrt[n]{\prod_{j=1}^n TPM_j(paternal)} = \frac{\exp\left(\widehat{\beta}\right)}{1 - \exp\left(\widehat{\beta}\right)}, \tag{46}$$

and

$$\sqrt[n]{\prod_{j=1}^n TPM_j(maternal)} = \frac{1}{1 - \exp\left(\widehat{\beta}\right)}. \tag{47}$$

For inferring an effect as statistically significant, we require that the mean posterior value of its corresponding $\delta$ parameter samples (which denote as posterior probability, PP) be higher than 0.95.

## Choice of hyper-parameter values and sensitivity analysis

As described above, our model has several hyper-parameters that need to be specified. Namely, the normal mixture prior distribution of each $\beta_i$ is scaled by respective $\tau_i$ and $c_i$ (*Equation 5*). The $c$ parameter acts as a multiplying constant that determines how much higher an important effect has to be relative to a negligible effect in order to be considered significant. The $\tau$ parameter allows scaling each effect independently. The shape and scale of the inverse gamma prior distribution of $\sigma^2$ is determined by $\nu$ and $\lambda$, respectively (*Equation 7*). The $\nu$ parameter should be set to a relative uninformative value, yet larger than zero in order to avoid obtaining low values of $\sigma^2$ which will result in often selecting effects as significant. Finally, each $\delta_i$ has prior probability $p_i$ of being selected.

We follow *Chipman et al. (1997)* in setting $\nu$ to a value near 2 (specifically 2.5) and setting $\lambda = \frac{2\sqrt{var(\widehat{y})}}{5} \frac{1}{\nu}\left(\frac{\nu}{2} - 1\right)$. Since $var(\widehat{y}) = 1$ for all transcripts, for $\nu = 2.5$ we get $\lambda = 0.04$. We chose $\tau$ and $c$

of all transcripts to have empirical values of 0.1 and 4.25, respectively, in order to provide a good separation between imprinted and non-imprinted transcripts at the selected PP = 0.95 cutoff. In addition, we set $p$ of all effects to 0.1, reflecting our prior belief of the effects being significant. Although the choice of these hyper-parameter values is arbitrary to some extent, if the model is robust this choice should not affect the ranking of transcripts according to the PPs. Therefore, in order to evaluate how the inference of genomic imprinting by our model is affected by our choice of empirical hyper-parameter values we performed the following sensitivity analysis. The empirical value that we chose for each of the five hyper-parameters: $\tau$, $c$, $\nu$, $\lambda$, and $p$, was perturbed by selecting four other values. Specifically, we perturbed the empirical $\tau = 0.1$ value with 0.005, 0.01, 1, and 2 thus both lowering and elevating the posterior probability of an effect being significant; we perturbed the empirical $c = 4.25$ value only with the higher values: 1.7, 3.4, 5.3125, and 10.625 as 4.25 was found to be approximately the minimal value for detecting significant effects; we perturbed the empirical $\nu = 2.5$ value with 5, 12.5, 25, and 50 as $\nu$ cannot assume values lower than 2; we perturbed the empirical $\lambda = 0.04$ value with 0.002, 0.004, 0.4, and 0.8 shifting the $\sigma^2$ prior distribution from informative to uninformative; and finally, we perturbed the empirical $p = 0.1$ value with higher prior probabilities of 0.2, 0.3, 0.4, and 0.5. In each such perturbation we re-fitted our model to the data where all other hyper-parameter values are held fixed at their original empirical unperturbed values (thereby achieving a one-at-a-time sensitivity analysis). We assessed the perturbed results using a receiver operating characteristic (ROC) analysis in which the imprinted transcripts obtained by the unperturbed inferences were used as the ground-truth positives, and all other non-imprinted transcripts were used as ground-truth negatives. For practical computational considerations, for all perturbations we used a random sample of 10% of the 47,676 transcripts in our data set. In all perturbations the area under the ROC curve (AUC) was found to be 1, except for the $\tau = 1$ and $\tau = 2$ perturbations which obtained AUCs of 0.99 and 0.95, respectively (*Figure 9*). These results therefore indicate that the ranking of transcripts according to the PP of the parental effect, obtained by our model, is robust to the choice of hyper-parameter empirical values. Therefore, the practice employed in this study of selecting empirical values, setting a parental effect PP cutoff, and experimentally confirming all transcripts with a PP of parental effect above that cutoff, is a reasonable choice for obtaining reliable inference of genomic imprinting from RNA-seq data.

In order to detect all imprinted transcripts in our data we fitted our model to all 48 P8 and P60 samples, thus estimating cross, sex, and age effects on the parental bias. This, however, requires that a given transcript is expressed in both age groups. In addition, a significant age effect may mean that a transcript has a strong parental bias in one age group but none in the other. In that case, the PP of the parental bias may not be significant. For these reasons we additionally fitted our model to the 24 P8 and P60 samples, independently, thus estimating only cross and sex effects on the parental bias for each of the age groups. We thus report all transcripts with a parental effect PP > 0.95 in the combined P8, P60 dataset, or exclusively expressed in one of the age group datasets with a parental effect PP > 0.95 as imprinted. Transcripts with an age effect PP > 0.95 in the P8, P60 dataset and with a parental effect PP > 0.95 in either of the individual P8 and P60 datasets were additionally reported as imprinted.

## Source code availability
BRAIM is implemented in R—see *Source code 1*.

## Analysis of the decay of parentally-biased expression as a function of distance from imprinted cluster centers
To quantify the effect that physical chromosomal distance of imprinted genes from their corresponding imprinted cluster center has on the magnitude of their parentally-biased expression we performed the following analysis. We started off with the list of transcripts which were either validated to be imprinted in this study or their corresponding genes were validated to be imprinted in previous studies (*Supplementary file 1H*). Retrogenes were excluded from this list as the context of their genomic location is not strongly relevant for the question addressed by this analysis. For each transcript in this list we computed the parental bias as:

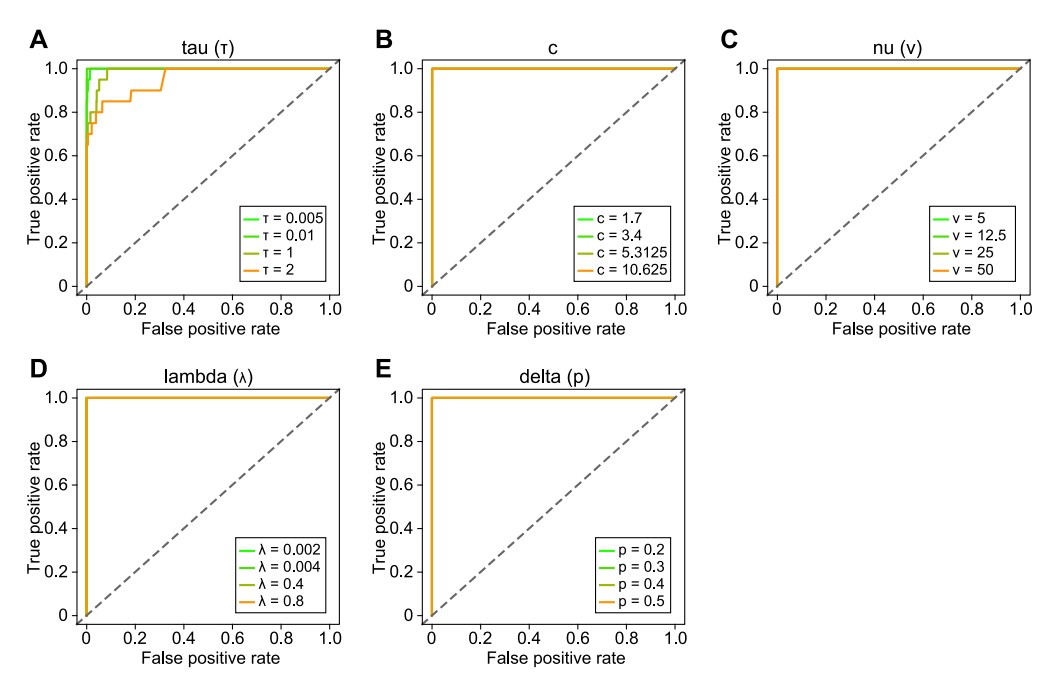

**Figure 9**. Sensitivity analysis of BRAIM to the choice of hyper-parameter values. Receiver operating characteristic curves describing the performance of BRAIM in detecting imprinted genes using perturbed values of $\tau$ (**A**), $c$ (**B**), $\nu$ (**C**), $\lambda$ (**D**), and the prior of $\delta$, $p$ (**E**) hyper-parameters.

$$\max\left(\frac{\exp(\widehat{\beta})}{1-\exp(\widehat{\beta})}, \frac{1}{1-\exp(\widehat{\beta})}\right). \tag{48}$$

In words, the median parental expression fraction across all samples (see *Equations 43–47*). In order to represent a gene by a single transcript, for each gene we chose the isoform with the most significant parental bias (i.e., the isoform with the maximal parental effect PP), derived from the model fitted to both age groups. The same criterion was also used to select between host and resident genes as the representative of their genomic location. In order to assign genes to clusters we linearly scanned each chromosome from its 5′ to 3′ end. The most upstream gene on a chromosome (where we represent the start and end sites of genes encoded on the antisense strand as their end and start sites on the sense strand, respectively), was assigned to the first imprinted cluster of that chromosome. Then, if the start site of the next downstream gene is within 1 MB of the end site of the previous upstream gene it was assigned to the same cluster, else it was assigned to a new cluster. Thus, the maximal distance between any two genes within a cluster can be 1 MB. We then defined the genes with a parental bias larger than 85:15% as candidate centers of their respective clusters. Since in some imprinted clusters more than a single gene met this condition, we grouped all physically consecutive candidate cluster center genes to a single gene which start site was defined as the start site of the most upstream candidate center gene and end site was defined as the end site of the most downstream candidate center gene. If an imprinted cluster resulted with more than one such group of candidate center genes flanked by genes with parental biases lower than 85:15% it was broken down to sub-clusters centered on each of these candidate centers. We defined the boundaries of each sub-cluster as half the distance between the end site of its center gene and the start site of the center gene of the adjacent sub-cluster (or up to 1 MB in the cases of the most up- and downstream sub-clusters) and thus flanking genes were assigned to the sub-cluster which boundary was downstream to their start site location. We finally removed any gene which start site

was more than 1 MB away from its respective cluster center. We then fitted a general linear fixed effects model to non-cluster center genes, where we defined the response as the parental bias and the fixed effect as the intergenic distance from the cluster center.

## Analysis of the evolutionary propensity for clustering of imprinted genes

To test whether the clustered organization of mouse imprinted genes with weak parental-expression biases (lower than 85:15%) is conserved in mammalian evolution, which therefore suggests functional importance, we carried out the following analysis of conservation of micro-synteny. We defined a pair of adjacent genes in the mouse reference genome as any two genes for which intergenic distance was below 1 MB (as 92% of the imprinted genes in our data are within 1 MB of another imprinted gene). Using this set of adjacent pairs of genes as reference we then constructed a phyletic pattern of orthologous gene pairs as follows. We downloaded the gene orthology assignment between the mouse reference genome and all mammalian reference genomes available in the Ensembl genome browser (*Flicek et al., 2014*), which are assembled at the chromosome level. This included the genomes of the following species: opossum, pig, sheep, cow, horse, cat, dog, rat, marmoset, rhesus, orangutan, gorilla, chimpanzee, and human (*Figure 5E*). For each comparative genome, if both orthologs of a mouse adjacent gene pair exist in that genome, we labeled this orthologous pair by '1' if their intergenic distance is lower than 1 MB (i.e., they are also adjacent in the comparative genome) and '0' otherwise (i.e., they are not adjacent in the comparative genome). If either or both orthologs do not exist in the comparative genome we labeled this pair as '?' (i.e., unknown). Ensembl tables of orthology assignment provide two types of homology between a query and search genomes: a one-to-one and a one-to-many homology. In addition, these tables also provide a binary confidence score for the orthology assignment (0 for low and 1 for high). As a conservative approach, we filtered out all one-to-many and low confidence orthologs unless the genes orthologous to the mouse pair were found to be within 1 MB of each other. As a result we obtained a phyletic pattern, which is a type of a multiple sequence alignment, where in our case each site in the phyletic pattern is a gene pair orthologous to an adjacent gene pair in the mouse genome with the characters of '1', '0', and '?'. All phyletic pattern columns in which all sites were '?', were removed thus obtaining a phyletic pattern of 409,874 sites. We then fitted to this phyletic pattern and corresponding phylogenetic species tree (*Figure 5E*) a probabilistic phylogenetic model estimating the substitution rates between presence and absence ('1' and '0', respectively) and vice-versa (*Cohen et al., 2008*). We specifically allowed the presence/absence ratio to vary across sites and the presence and absence probabilities at the root of the phylogenetic tree to be independently estimated. As a result, we obtained a posterior expectation of the absence-to-presence substitution rate (i.e., rate of gene-pair gain: '0' to '1') and a posterior expectation of the presence-to-absence substitution rate (i.e., rate of gene-pair loss: '1' to '0') for each site in the phyletic pattern. The propensity of each orthologous gene pair to be adjacent throughout the mammalian evolution represented by the species used in this analysis is therefore the posterior expectation of its rate of gene-pair gain divided by the posterior expectation of its rate of gene-pair loss. In order to test whether the propensity of adjacency of orthologs of mouse weakly parentally-biased imprinted gene pairs (with a bias lower than 85:15%), is significantly higher than what would be expected by chance, we compared the mean propensity of adjacency of these gene pairs with the mean propensity of adjacency of all gene pairs in the phyletic pattern using a one-sided *z*-test. We note that not applying any distance cutoff for defining a pair of adjacent genes means that genes with much longer intergenic distances will be included for which the micro-synteny is expected to be much less evolutionary conserved thus potentially biasing our analysis.

## Pyrosequencing validations

Pyrosequencing validations were performed in cerebella derived from a different batch of F1 hybrids from that used in the RNA-seq experiments. The same age of animals was used for the pyrosequencing validations whenever the RNA-seq data showed an age-specific parental bias. Otherwise, either age group was used. RNA purification and quality control followed the same procedures described for the RNA-seq data. An average of three SNPs suitable for pyrosequencing analyses were identified for each candidate gene. Previously described three-primer strategy (*Royo et al., 2007*), including a 3′-biotinylated primer common to all reactions, was employed for the amplification of each targeted SNP. All primers were designed using Pyromark Assay Design 2.0. Pyromark One Step RT-PCR kit (Qiagen) was used for the amplification of each targeted region,

followed by purification of single stranded biotinylated DNA according to the manufacturer instructions. Pyrosequencing was performed on the Pyromark Q96 MD (Qiagen). For each SNP at least 12 replicates were separately analyzed. Statistical analyses of the pyrosequencing data for each SNP were performed using BRAIM.

## Pyrosequencing analyses of spatial and temporal regulation

For each tissue we collected six samples from F1i and F1r females. The coordinates of the Allen Reference Atlas guided dissections of all brain regions interrogated. RNA purification, pyrosequencing and analyses of parental biases were conducted as described for pyrosequencing validations. For the heat maps in *Figure 6D*, *Figure 6—figure supplement 1*, and *Figure 7B* we binned the parental biases with PP > 0.95 to five bins: 50:50–60:40, 60:40–70:30, 70:30–80:20, 80:20–90:10, and 90:10–100:0, and assigned the parental biases with PP ≤ 0.95 to the 50:50 category. The bars in *Figure 6C*, *Figure 6—figure supplement 2A,B*, *Figure 7A*, and *Figure 8B,C*, correspond to the actual parental biases, regardless of their PPs, along with an error notch corresponding to the 95% confidence interval (*Supplementary file 1H, I*).

To quantify the statistical significance of the difference in the magnitude of the parental bias of each gene between the 16 brain regions and 7 non-neural somatic tissues we used a *t*-test (assuming unequal variances) comparing its brain and non-neural somatic tissues samples. To quantify the statistical significance of the variability of the parental biases across the different brain regions, for each gene, we used a one-way ANOVA test, where the response is the parental bias computed for all samples and the independent variable is the brain region.

## Generation of mice and preparation of tissue for the analysis of *Bcl-x*

Mice with a nervous system-specific paternal deletion of *Bcl-x* were generated by crossing males heterozygous for both a *Nestin::Cre* transgene (*Tronche et al., 1999*) and a floxed allele of *Bcl-x* (*Rucker et al., 2000*) with WT females (carrying two WT alleles of *Bcl-x* and lacking the *Nestin::Cre* transgene) that had a strain background that was a 50% mixture of each of the *Nestin::Cre* (+/+) and *Bcl-x* (fl/fl) background strains. Mice with a nervous system-specific maternal deletion of *Bcl-x* were generated by crossing females bearing a floxed allele of *Bcl-x* with males bearing a *Nestin::Cre* transgene. From the same crosses, mice bearing the *Nestin::Cre* transgene and carrying two WT alleles of *Bcl-x* were used as WT littermate controls. This is expected to knock out the function of both $Bcl-x_L$ and $Bcl-x_S$, as both proteins are encoded primarily by exon 2, however, $Bcl-x_L$ is believed to be substantially more expressed in the brain (*Krajewska et al., 2002*), and thus primarily affected by the *Nestin::Cre* driven recombination. To minimize any possibility of background strain effects, all mice analyzed had a strain background that was a 50% mixture of each of the *Nestin::Cre* (+/+) and *Bcl-x* (fl/fl) background strains. Genotyping was performed by PCR, as described (*Rucker et al., 2000*).

A non-experimenter performed the genotyping. This person assigned unique animal numbers to each of the animals such that from here-on-out, researchers were completely blinded to the genotypes. Researchers performed all of the tissue-processing and image quantifications fully-blinded to the genotypes of the animals. The data were then given to another researcher who unblinded the animals' genotype, performed statistics, and created figures.

Mice aged P78-85 were weighed, transcardially perfused with PBS followed by 4% paraformaldehyde (PFA) in PBS. Brains were removed, weighed, postfixed in 4% PFA (in PBS) for 4 hr at 4°C, immersed in 30% sucrose (in PBS) overnight at 4°C, and then frozen at −80°C until being sectioned. Coronal sections (14 μm) of brain from male mice were prepared using a sliding microtome (Leica, Germany) and mounted in series on eight slides, which were subsequently stored at −80°C.

## NISSL staining

Area quantifications were performed on brain sections stained using the NISSL method. In brief, sections were hydrated in a graded series of alcohol, stained with cresyl violet, dehydrated in alcohol, cleared with xylenes, and coverslipped with DPX.

## Two-color immunofluorescence (IF)

Slide-mounted sections were warmed (37°C, 5 min), equilibrated in PBS (5 min, RT), fixed in PFA (4% in PBS; 10 min, RT), washed in PBS (3 min, RT), permeabilized with Triton X-100 (0.5% in PBS; 30 min, RT),

washed in TNT (3 × 5 min, RT), blocked in fetal bovine serum (FBS; 10% in TN; 30 min, RT), incubated with mouse anti-NEUN (EMD Millipore, Billerica, MA, MAB377; 1:1000) and rabbit anti-S100ß (Abcam, UK, ab41548; 1:500) antibodies (in 10% FBS; 12 hr, 4˚C), washed in TNT (3 × 5 min, RT), incubated with secondary antibodies (Alexa488- and Alexa647-labeled; Invitrogen, Carlsbad, CA; 1:1000 in 10% FBS; 12 hr, 4˚C), and washed in TNT (3 × 15 min, RT). Slides were mounted using Vectashield containing DAPI (5 µg/ml).

## Two-color fluorescent in situ hybridization

Probe target sequences for *Vglut1* (*Slc17a7*) and *Gad1* were amplified by PCR and inserted into pCRII-TOPO vector (Life Technologies). Sequences were identical to those used by the Allen Brain Atlas (http://www.brain-map.org/). Labeled antisense RNA probes were generated from linearized plasmid template using T7 or Sp6 polymerases (Promega, Madison, WI) with a digoxigenin RNA labeling mix (Roche, Switzerland; for *Gad1*) or a dinitrophenyl RNA labeling mix (Perkin Elmer, Waltham, MA; for *Vglut1*), treated with DNaseI (Promega), ethanol precipitated, and dissolved in a 30-µl volume of water.

Slide-mounted sections were warmed (37˚C, 5 min), equilibrated in phosphate-buffered saline (PBS; pH 7.2; 5 min, room temperature [RT]), fixed in PFA (4% in PBS; 10 min, RT), washed in PBS (3 min, RT), permeabilized with Triton-X-100 (0.5% in PBS; 10 min, RT) followed by Proteinase K (20 µg/ml in 50 mM Tris, 5 mM EDTA; 15 min, RT with gentle shaking), washed in PBS (3 × 3 min, RT), fixed in PFA (4% in PBS; 10 min, RT), washed in PBS (3 × 3 min, RT), incubated in acetylation solution (triethanolamine [0.1 M; pH 7.5], acetic anhydride [0.25%]; 10 min, RT), washed in PBS (3 × 3 min, RT), incubated in hybridization solution (formamide [50%], SSC [5×], Denhardts [5×], yeast tRNA [250 µg/ml], herring sperm DNA [200 µg/ml]; 30 min, RT), hybridized simultaneously with both *Vglut1* and *Gad1* antisense RNA probes (1:300 each in hybridization solution; 16 hr, 68˚C), washed with SSC (2×; 5 min, 68˚C), washed with SSC (0.2×; 3 × 30 min, 68˚C), incubated in $H_2O_2$ (3% in TN [Tris-HCl (0.1 M; pH 7.5), 0.15 M NaCl]; 30 min, RT), washed in TNT (Tween-20 [0.05%] in TN; 3 × 3 min, RT), incubated in TNB (Blocking Reagent [Perkin Elmer; 0.05% in TN]; 30 min, RT), incubated with anti-digoxigenin-POD antibody (1:1000 in TNB; 12 hr, 4˚C), and washed in TNT (3 × 20 min, RT). Fluorescent signals corresponding to the *Gad1* probe were generated using the Tyramide Signal Amplification (TSA) Plus Fluorescein Kit (Perkin Elmer) according to the manufacturer's instructions, after which sections were washed in TNT (2 × 3 min, RT), incubated in $H_2O_2$ (3% in TN; 1 hr, RT), washed in TNT (3 × 3 min, RT), incubated with anti-dinitrophenyl-HRP antibody (Perkin Elmer; 1:500 in TNB; 12 hr at 4˚C), and washed in TNT (3 × 20 min, RT). Fluorescent signals corresponding to the *Vglut1* probe were generated using the TSA Plus Cyanine5 Kit (Perkin Elmer) according to the manufacturer's instructions. Slides were mounted using Vectashield (Vector Laboratories) containing DAPI (5 µg/ml).

## Image capture, analysis and quantification

Sections were imaged using a Zeiss (Germany) Axioscan.Z1 microscope with a 10× objective. Areas of brain regions were measured from images of NISSL-stained sections using Zen software (Zeiss). Cell densities and numbers were quantified from two-color IF and in situ hybridization images using ImageJ software.

Five anterior cortical sections that include the somatomotor cortex (corresponding to Figures 29–33 in *Paxinos and Franklin, 2007*) were analyzed. Three coronal cerebellar sections that included lobules 4–6 of the cerebellar vermis (corresponding to Figures 86–88 in *Paxinos and Franklin, 2007*) were analyzed. Four olfactory bulb sections immediately rostral to the external plexiform layer of the accessory olfactory bulb (including and rostral to Figure 2 in *Paxinos and Franklin, 2007*) were analyzed.

## Accession numbers

The RNA-seq raw and processed data as well as *Supplementary file 1* have been deposited to GEO database under the accession number GSE67556.

## Acknowledgements

We thank Dr Tirthankar Dasgupta, (Department of Statistics, Harvard University), Dr Cole Trapnell (Department of Genome Sciences, University of Washington), and Dr Ming Hu (Department of Population Health, New York University) for helpful advice and discussions. We are grateful to Stacey

Sullivan for help with *Bcl-x* mutant mice. We thank the staff of Harvard FAS Center for System Biology, Resources and Instrumentation, the staff of Harvard University FAS Research Computing, and the staff of Harvard Center for Biological Imaging for assistance and advice. We are thankful to Renate Hellmiss for invaluable assistance with graphical presentation. We also thank Dr Christopher Gregg for advice and helpful discussions at the onset of the project. Finally, we thank Dr Yoh Isogai, Dr Anne Lanjuin, Dr Joseph Bergan, and other members of the Dulac laboratory for their frequent advice and unconditional support. This work was funded by the Howard Hughes Medical Institute (HHMI) and the NIMH's Silvio O. Conte Center Grant (P50MH094271) for Basic and Translational Research.

## Additional information

### Competing interests
CD: Senior editor, *eLife*. The other authors declare that no competing interests exist.

### Funding

| Funder | Grant reference | Author |
| --- | --- | --- |
| Howard Hughes Medical Institute (HHMI) | NA | Catherine Dulac |
| National Institute of Mental Health (NIMH) | 1P50MH094271 | Catherine Dulac |

The funders had no role in study design, data collection and interpretation, or the decision to submit the work for publication.

### Author contributions
JDP, NDR, Conception and design, Acquisition of data, Analysis and interpretation of data, Drafting or revising the article, Contributed unpublished essential data or reagents; DEF, Analysis and interpretation of data; SWS, LAN, Acquisition of data, Analysis and interpretation of data, Drafting or revising the article; OH-S, Analysis and interpretation of data, Contributed unpublished essential data or reagents; JJC, Acquisition of data; MZ, S-KC, Acquisition of data, Analysis and interpretation of data; JSL, Conception and design, Analysis and interpretation of data; CD, Conception and design, Analysis and interpretation of data, Drafting or revising the article, Contributed unpublished essential data or reagents

### Ethics
Animal experimentation: This study was performed within the facilities of the Harvard University Faculty of Arts and Sciences (HU/FAS) in strict accordance with the recommendations in the Guide for the Care and Use of Laboratory Animals of the National Institutes of Health. All animals were handled according to a protocol approved by the Harvard University Institutional Animal Care and Use Committee (IACUC; protocol #97-03). The HU/FAS animal care and use program maintains full AAALAC accreditation, is assured with OLAW (A3593-01), and is currently registered with the USDA. Every effort was made to minimize animal suffering during this study.

## Additional files

### Supplementary files
• Supplementary file 1. (**A**) Results of BRAIM Analysis of the Autosomal Transcriptomes in the P8 and P60 Cerebellum. (**B**) Results of BRAIM Analysis of the Autosomal Transcriptome in the P8 Cerebellum. (**C**) Results of BRAIM Analysis of the Autosomal Transcriptome in the P60 Cerebellum. (**D**) Results of BRAIM Analysis of X-linked Transcriptomes in the P8 and P60 Cerebellum. (**E**) Results of BRAIM Analysis of X-linked Transcriptome in the P8 Cerebellum. (**F**) Results of BRAIM Analysis of X-linked Transcriptome in the P60 Cerebellum. (**G**) List of Imprinted Genes Identified in the Mouse Cerebellum. (**H**) Curated List of Genes Reported in the Literature to be Imprinted in the Mouse. (**I**) Results of Pyrosequencing Validation of BRAIM-Reported Imprinted Genes in the Mouse Cerebellum. (**J**) Pyrosequencing Quantification of Parental Biases of Imprinted Genes Across Multiple Brain Regions and Body Tissues. (**K**) Comparison of Parental Biases of Imprinted Genes Between Brain and Body

Tissues. (**L**) Analysis of the Variance of Parental Biases Across Brain Regions. (**M**) Pyrosequencing Quantification of Parental Biases of Imprinted Genes From Multiple Brain Regions at Different Developmental Stages.

• **Source code 1.** The compressed source code folder, braimSourceCode.zip, includes the BRAIM R code along with example parameters and output and a help readme file which explains how to run the code.

### Major dataset

The following dataset was generated:

| Author(s) | Year | Dataset title | Dataset ID and/or URL | Database, license, and accessibility information |
|---|---|---|---|---|
| Perez JD, Rubinstein ND, Fernandez DE, Dulac C | 2015 | Parent-of-Origin Allelic Expression in the Mouse Cerebellum | http://www.ncbi.nlm.nih.gov/geo/query/acc.cgi?acc=GSE67556 | Publicly available at the NCBI Gene Expression Omnibus (Accession no: GSE67556). |

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
