## [Decision Letter]

Thank you for submitting your work entitled “Regulation and Functional Significance of Parent-of-Origin Allelic Expression in the Brain” for peer review at *eLife*. Your submission has been favorably evaluated by Eve Marder (Senior editor), Sacha Nelson (Reviewing editor and reviewer), and Sean Eddy (reviewer).

The reviewers have discussed the reviews with one another and the Reviewing editor has drafted this decision to help you prepare a revised submission.

As you can see both reviewers are largely favorable but together have numerous suggestions for improving the manuscript through textual changes, clarifications or in some cases additional or revised analyses. Since none of these require additional experiments they are conveyed in full rather than in synthesized form. It is possible that some of the points raised can be dealt with through clarifications to the reviewers and that additional editorial discussion may be required at that time.

*Reviewer #1 (Reviewing editor)*:

Substantive concerns:

This is an extremely thorough analysis of imprinting in the brain which follows up on several less carefully controlled studies including one from the senior author's lab. The analysis is rigorous, the follow up validation experiments are convincing and the results with respect to control of developmentally regulated apoptosis and brain size are interesting and important.

I have only one concern which might be termed major:

Figure 8 illustrates an analysis of imprinting by pyrosequencing across 16 brain regions. I find the similarity across brain regions more impressive for most genes tested than the differences. The differences between brain and peripheral tissues are, on the other hand, striking. I think the authors should tone down their discussion of this point (e.g. perhaps the variation across brain regions should be called “modest but significant” – assuming there is some test of significance, instead of “substantial”; in the subsection “Dynamic Temporal and Spatial Regulation of Imprinted Genes in the Brain”) Alternatively, and/or additionally, the authors could provide some more quantitative analysis that supports the idea that it is a “major” effect (e.g. relative to age or brain vs. periphery).

Minor comments [abridged]:

In the subsection “Developmental Regulation of Genomic Imprinting in the Cerebellum”, 42 of the 115 imprinted genes are differentially expressed between P8 and P60 ([Supplementary-material SD1-data]–Table 1). This is a significantly higher proportion than the 6,531 out of 25,058 genes expressed in the cerebellum that are regulated according to developmental stage (P-value =3.8∼10-8; hypergeometric test). 25K genes seems very high for the number of genes expressed. Is this transcripts? Is it the case that essentially all genes are expressed in the cerebellum? Or is this an error?

*Reviewer #2*:

Substantive concerns:

The authors describe an RNA-seq screen for new imprinted genes in the mouse cerebellum. The screen extends and improved upon previous work by using an experimental design with many replicants; by using a sophisticated statistical analysis that accounts for other main sources of biological variation; and by systematically following up each candidate imprinted gene from the RNA-seq analysis with a second validation assay using specific allele-specific pyroseq analysis.

The authors discover some new imprinted genes and they also make three particularly interesting observations: (1) that the degree of paternal bias for a given locus is quantitative, with some loci being only weakly or moderately biased, whereas imprinted loci are typically discussed as if they are all-or-nothing, monoallelically expressed; (2) that at least to some extent, this quantitative bias can be accounted for by a gradient away from a strong imprinting center, with loci progressively further from that point showing progressively less strong bias; and (3) that the strength and even the direction of the bias can vary in a cell type specific manner. I am not an expert in imprinted genes, but these observations seem very interesting to me, and they greatly extend my understanding of the phenomenology of parent-of-origin effects.

There is a lot of evidence of care and rigor in the manuscript and its experiments. This includes the technically sophisticated statistical model, which appears to have been carefully done and carefully tested (for example, with a parameter sensitivity analysis).

The paper describes the results of a screen, so its contribution is necessarily more descriptive – laying ground for future work – than getting at mechanisms. That said, in some places, it would strengthen the paper to have more discussion of possible mechanisms of imprinting, even if speculative, to help the reader organize the mass of descriptive observations, which can tend to become a blur.

I understand (as an outsider to the field) that there is a somewhat controversial history of previous work in this area. I think it would help, and would even strengthen the paper, to clarify that history in the Introduction. Specifically, the [35] paper was particularly controversial (perhaps among others?) because it claimed to have detected over 1000 imprinted genes, when only around 100 were known. The [22] paper made a convincing case that the [35] result was in error, a large overestimation as a result of deficiencies in statistical analysis and validation of RNA-seq information. This manuscript seems to agree with that conclusion, but the Introduction seems cagily written: [35] and [22] are cited simultaneously in a way that makes it seem like they both pointed to challenges in this sort of screen, rather than as [35] making errors that [22] needed to walk back. For example, when I first read “… methodologically challenging (22; 35)”, I assumed that both DeVeale and Gregg were papers about the analysis difficulties. It was not clear to me, until I looked into it more, that Gregg was a high profile result that DeVeale challenged. I think it would be brave, and helpful to the field of RNA-seq statistical analysis, to be even more candid about this history.

Minor comments [abridged]:

Figure 1. I'm confused by the distribution of points on this plot. Consider the orange point at roughly 0.8, 0.8. This is called a “false positive”, with a preferred allele frequency of 0.8 as measured both by RNA-seq and by pyroseq validation. Why is this a “false positive”, if pyroseq validation agrees with the allele frequency measured by RNA-seq? From your explanation I expected that the only way something can be a “false positive” is if it gets called imprinted from RNA-seq data, but fails to validate with pyroseq data. (Also, Figure 1 appears to show 10 false positive points, whereas Figure 1 says there are 9 false positives.)

Figure 1—figure supplement 1. I am worried that an error has entered here. The Methods states that “transcripts with low expression values need to be filtered”, and “a typical distribution of TPM values is bimodal where the left mode represents transcripts with low expression”, and I agree. But the plot shown in Figure 1—figure supplement 1 is trimodal, not bimodal. The bimodality of expression levels in RNA-seq data is usually seen with modes around TPM ∼0.01-0.1 (“off” genes) and 10-1000 (“on” genes). I think these are your rightmost two modes. The third (leftmost) mode at log(TPM)∼-10 is, I'm guessing, a software artifact of MM-SEQ estimation of expression levels given little or no mapped reads. It seems unlikely that you have any data for a transcript at TPM = 10^-10. I think you should be drawing your cutoff threshold at around log(TPM)∼0, between the rightmost two modes. One place this affects you is in the top panel of Figure 2, *Fkbp6*: I think this gene is basically off (TPM ∼ 0.1 or so), and accordingly it's showing very high variance.

Figure 2: Individual data points are shown with error bars. Define what these error bars are. These data are individual measurements from single samples, so there is no variance derived from replications that you could be plotting (I think). I imagine that the error bars are related to uncertainties reported by MM-SEQ for its estimated TPM calls, based on the counting error given the number of mapped reads it had. If it's a posterior distribution, put the range in the legend (95% posterior interval?).

In the subsection “Transcriptome-Wide Profile of Parent-of-Origin Expression in the Mouse Cerebellum”, you state: “our approach has a true positive rate of 93%” – based on what? Perhaps I missed it but I don't think you ever say what you consider to be the “true” number, or how you could know it. You could mean that ∼80 genes have been previously reported as imprinted, 74 of which you detected (74/80 = 0.93), but surely more than 80 genes have previously been reported and validated as imprinted? I thought the number was >100. “We also considered the possibility that true imprinted genes may not meet our parental bias PP cutoff”: The “also” is confusing because in the previous paragraph you've just said that your true positive rate (sensitivity) is 93% – which already tells you that the false negative rate – trues that you miss – should be 7%. But in this paragraph you find a false negative rate of 10/18 = 56%, so that isn't consistent with your reported true positive rate. I suppose the key is in what you mean by “subjectively selected”. You must have selected genes that you expect to be imprinted, rather than random negatives. Clarify. For example, you might lead with something like: “The fact that our true positive rate is <100% means that we are sure to miss some genes that are imprinted…”

In the subsection “Developmental Regulation of Genomic Imprinting in the Cerebellum”, the sentence: “Such a pattern may artificially arise if the ability to detect the parental bias increases with its overall expression level. However, this does not appear to be the case in our data, as parental biases are accurately estimated across a wide spectrum of expression levels (Figure 4—figure supplement 1)”. It seems to me that it must be the case that the ability to detect parental bias increases with overall expression level. That seems unavoidable. And I don't see how the data in the supplement argue against it. Neither Figure 4—figure supplement 1 shows any data directly relevant to the “accuracy” of parental bias estimation. Figure 4—figure supplement 1 shows a dearth of points in the low expression/weak bias quadrant, as expected if ability to detect bias increases with expression. Figure 4—figure supplement 1 also shows a weird dearth of points in the low expression/*high* bias quadrant. Explain? (Is that just the fact that the Bayesian estimation of parental bias is pulled back toward 0.5 by the prior, in cases of very low expression with little data?).

In the subsection “Transcriptome-Wide Analysis of Genomic Imprinting in the Cerebellum”, the sentence: “In addition to 74 imprinted genes previously validated… our analysis detects 41 novel imprinted genes… increasing the total number of reported imprinted genes in the mouse by 26%”. Unclear how the 26% number is arrived at; as with the comment about true positive rate, I don't think you've stated what you think the total number of reported imprinted genes is. This needs to be stated clearly, especially since [34] reported >1000, but was overestimating.

I don't think the content of the paper has much to do with the “functional significance” of parental bias, nor does it have much to do with its “regulation” except in the descriptive sense that you've observed differences. The Title of the manuscript could just be “Parent of Origin Allelic Expression in the Brain”.

---

## [Author Response]

Reviewer #1 (Reviewing editor):

*Substantive concerns*:

Figure 8
*illustrates an analysis of imprinting by pyrosequencing across 16 brain regions. I find the similarity across brain regions more impressive for most genes tested than the differences. The differences between brain and peripheral tissues are, on the other, hand striking. I think the authors should tone down their discussion of this point (e.g. perhaps the variation across brain regions should be called “modest but significant” – assuming there is some test of significance, instead of “substantial”; in the subsection “Dynamic Temporal and Spatial Regulation of Imprinted Genes in the Brain”) Alternatively, and/or additionally, the authors could provide some more quantitative analysis that supports the idea that it is a “major” effect (e.g. relative to age or brain vs. periphery)*.

We thank the reviewer for bringing up this important point and for the helpful suggestions. The variability should indeed have been better assessed statistically. We have now addressed these points by quantifying the brain vs. non-neural tissues difference in the magnitude of parental bias as well as by quantifying the variability among-brain-regions in the magnitude of parental bias. The former shows a significant difference for nearly all tested genes. The latter test also reports significant variability for a large fraction of genes (18 out of the 28 tested). These tests are referred to in the text and their results are added to Figure 6. We also updated the text in the Discussion accordingly.

*Minor comments [abridged]*:

*In the subsection “Developmental Regulation of Genomic Imprinting in the Cerebellum”, 42 of the 115 imprinted genes are differentially expressed between P8 and P60 (*[Supplementary-material SD1-data]*–Table 1). This is a significantly higher proportion than the 6,531 out of 25,058 genes expressed in the cerebellum that are regulated according to developmental stage (P-value =3.8∼10-8; hypergeometric test). 25K genes seems very high for the number of genes expressed. Is this transcripts? Is it the case that essentially all genes are expressed in the cerebellum? Or is this an error*?

We thank the reviewer for pointing out this issue. Our sequencing protocol captures both coding and noncoding RNAs. In our analyses we did not limit ourselves only to coding RNAs, and the noncoding RNAs make up a large fraction of the number of transcripts we report. We have now written this more explicitly (please see the subsection “Transcriptome-Wide Profile of Parent-of-Origin Expression in the Mouse Cerebellum”) specifying that among the 49,464 heterozygous autosomal and X-linked transcripts for which we tested for parentally biased expression, 32,399 transcripts are protein coding and 17,065 are non-coding. This corresponds to 16,841 protein coding and 14,706 non-coding genes. In the Materials and methods section, under “Reference Genome and Transcriptome Data”, we now describe the composition of the different types of non-coding transcripts we used.

Reviewer #2:

*Substantive concerns*:

*The paper describes the results of a screen, so its contribution is necessarily more descriptive – laying ground for future work – than getting at mechanisms. That said, in some places, it would strengthen the paper to have more discussion of possible mechanisms of imprinting, even if speculative, to help the reader organize the mass of descriptive observations, which can tend to become a blur*.

*I understand (as an outsider to the field) that there is a somewhat controversial history of previous work in this area. I think it would help, and would even strengthen the paper, to clarify that history in the Introduction. Specifically, the*
[35]
*paper was particularly controversial (perhaps among others?) because it claimed to have detected over 1000 imprinted genes, when only around 100 were known. The*
[22]
*paper made a convincing case that the*
[35]
*result was in error, a large overestimation as a result of deficiencies in statistical analysis and validation of RNA-seq information. This manuscript seems to agree with that conclusion, but the Introduction seems cagily written:*
[35]
*and*
[22]
*are cited simultaneously in a way that makes it seem like they both pointed to challenges in this sort of screen, rather than as*
[35]
*making errors that*
[22]
*needed to walk back. For example, when I first read “… methodologically challenging (*[22]*;*
[35]*)”, I assumed that both DeVeale and Gregg were papers about the analysis difficulties. It was not clear to me, until I looked into it more, that Gregg was a high profile result that DeVeale challenged. I think it would be brave, and helpful to the field of RNA-seq statistical analysis, to be even more candid about this history*.

We thank the reviewer for pointing this out. We changed the text to reflect more candidly the course of events so that it is accessible to readers who are not familiar with that literature (Introduction). We believe that the narrative in our updated text is a fair description of the strengths and weaknesses of the earlier approaches, including our own.

*Minor comments [abridged]*:

Figure 1*. I'm confused by the distribution of points on this plot. Consider the orange point at roughly 0.8, 0.8. This is called a “false positive”, with a preferred allele frequency of 0.8 as measured both by RNA-seq and by pyroseq validation. Why is this a “false positive”, if pyroseq validation agrees with the allele frequency measured by RNA-seq? From your explanation I expected that the only way something can be a “false positive” is if it gets called imprinted from RNA-seq data, but fails to validate with pyroseq data. (Also,*
Figure 1
*appears to show 10 false positive points, whereas*
Figure 1
*says there are 9 false positives*.*)*

We thank the reviewer for pointing out this confusing point. Indeed several details of Figure 1 require clarification. The orange dot with coordinates at ∼0.8,0.8 was considered a false positive because the preferred allele observed in the RNA-seq data is opposite to the one observed by pyrosequencing. This dot represents one of the transcripts of the *Slc22a1* gene that on the RNA-seq platform shows that approximately 80% of its expression comes from the maternal allele, which in spite of substantial variability among biological samples has a marginal posterior probability above 0.95. However, pyrosequencing analysis of this same transcript reports an average of approximately 80% of its expression from the paternal allele with high variability and this effect is not significant (PP = 0.86). A similar situation occurred for *Cpa4*, which is represented by the orange dot at ∼0.6,0.8. The RNA-seq data show that approximately 60% of its expression comes from the maternal allele, whereas in pyrosequencing a more substantial 80% maternal expression is observed, yet due to the high variability this effect is not significant (PP = 0.77). Since these two genes are the only examples of this situation, it is clearly an exception in our analyses. We updated the legend of Figure 1 to clarify this point.

Figure 1—figure supplement 1*. I am worried that an error has entered here. The Methods states that “transcripts with low expression values need to be filtered”, and “a typical distribution of TPM values is bimodal where the left mode represents transcripts with low expression”, and I agree. But the plot shown in*
Figure 1—figure supplement 1
*is trimodal, not bimodal. The bimodality of expression levels in RNA-seq data is usually seen with modes around TPM ∼0.01-0.1 (“off” genes) and 10-1000 (“on” genes). I think these are your rightmost two modes. The third (leftmost) mode at log(TPM)∼-10 is, I'm guessing, a software artifact of MM-SEQ estimation of expression levels given little or no mapped reads. It seems unlikely that you have any data for a transcript at TPM = 10^-10. I think you should be drawing your cutoff threshold at around log(TPM)∼0, between the rightmost two modes. One place this affects you is in the top panel of*
Figure 2*,* Fkbp6*: I think this gene is basically off (TPM ∼ 0.1 or so), and accordingly it's showing very high variance*.

We are greatly thankful to the reviewer for bringing up this point. The distribution of ln(TPM) in a single RNA-seq sample is indeed trimodal, and the left mode is indeed what MMSEQ reports, where for transcripts whose expression levels are nearly, but not absolutely, zero MMSEQ reports posterior FPKM samples as low as 10^-07^. If MMSEQ’s posterior FPKM variances are taken to represent the expression-level uncertainty (as we did in our model) this results with false negatives in cases where a silenced or nearly silenced allele is in the left mode but due to this artificially high uncertainty the parental bias does not come out as significant. We are aware of this limitation and all transcripts whose mean posterior FPKM (which we convert to TPM units) fell below our cutoff were set to zero. We subsequently add the cutoff value to all posterior TPM samples thereby avoid taking the log of zero. We had an error in the figure and associated text indicating the cutoff is determined as the local minimum between the left two modes for each RNA-seq sample. We use a fixed cutoff of TPM = 0.01 for all RNA-seq samples and Figure 1—figure supplement 1 is now updated and shows the mean ln(TPM) values of all expressed transcripts (not only heterozygous) across all our RNA-seq samples.

We feel that selecting an expression cutoff of ln(TPM) = 0 is too strict for this analysis as many imprinted genes expressed below that level, that we carefully validated, have features of known and novel-validated imprinted genes expressed above that level, e.g., located in imprinted clusters, isoform-specific imprinting, temporal and spatial regulation, and associated with apoptosis. Using a more stringent expression level cutoff, which we determined empirically and describe in the Materials and methods (represented by the dashed line in Figure 1—figure supplement 1 and the newly added 1C), has a negligible effect on our results.

Figure 2*: Individual data points are shown with error bars. Define what these error bars are. These data are individual measurements from single samples, so there is no variance derived from replications that you could be plotting (I think). I imagine that the error bars are related to uncertainties reported by MM-SEQ for its estimated TPM calls, based on the counting error given the number of mapped reads it had. If it's a posterior distribution, put the range in the legend (95% posterior interval?)*.

We thank the reviewer for pointing this out and apologize for only describing this in the Materials and methods.

We quantified expression with MMSEQ, which reports a posterior distribution of the FPKM values (which, as we describe above, convert to TPM units and then take the natural log of). The midline of each box in the figure is the posterior mean ln(TPM), the box stretches one posterior ln(TPM) standard deviation from the posterior mean, and the notches at the bottom and top are the minimum and maximum posterior samples, respectively. We now describe this is in the legend of Figure 2.

*In the subsection “Transcriptome-Wide Profile of Parent-of-Origin Expression in the Mouse Cerebellum”, you state: “our approach has a true positive rate of 93%” – based on what? Perhaps I missed it but I don't think you ever say what you consider to be the “true” number, or how you could know it. You could mean that ∼80 genes have been previously reported as imprinted, 74 of which you detected (74/80 = 0.93), but surely more than 80 genes have previously been reported and validated as imprinted? I thought the number was >100. “We also considered the possibility that true imprinted genes may not meet our parental bias PP cutoff”: The “also” is confusing because in the previous paragraph you've just said that your true positive rate (sensitivity) is 93% – which already tells you that the false negative rate – trues that you miss – should be 7%. But in this paragraph you find a false negative rate of 10/18 = 56%, so that isn't consistent with your reported true positive rate. I suppose the key is in what you mean by “subjectively selected”. You must have selected genes that you expect to be imprinted, rather than random negatives. Clarify. For example, you might lead with something like: “The fact that our true positive rate is <100% means that we are sure to miss some genes that are imprinted…*”

We thank the reviewer for these two points. We consider all genes that we detected as parentally biased (above our marginal posterior probability cutoff) that were previously reported as imprinted and independently validated (e.g., detected in an RNA-seq screen and independently validated using pyrosequencing) as true positives. This amounts to 74 genes. We additionally consider as true positives the genes that we detected as parentally biased and validated using pyrosequencing that were either not previously reported as imprinted or not independently validated. This amounts to 41 genes out of the 50 genes that we detected as parentally biased. Hence, our precision (which we confused with true positive rate) is calculated as: (74+41)/(74+50) = 92.7%. We now write this clearly in the subsection “Transcriptome-Wide Profile of Parent-of-Origin Expression in the Mouse Cerebellum”. We do not rely on the 10/18 genes that we validated below our cutoff to represent the sum of false negatives (there may be more), hence we cannot compute the true positive rate. This is now clarified in the text.

*In the subsection “Developmental Regulation of Genomic Imprinting in the Cerebellum”, the sentence: “Such a pattern may artificially arise if the ability to detect the parental bias increases with its overall expression level. However, this does not appear to be the case in our data, as parental biases are accurately estimated across a wide spectrum of expression levels (*Figure 4—figure supplement 1*)”. It seems to me that it must be the case that the ability to detect parental bias increases with overall expression level. That seems unavoidable. And I don't see how the data in the supplement argue against it. Neither*
Figure 4—figure supplement 1
*shows any data directly relevant to the “accuracy” of parental bias estimation.*
Figure 4—figure supplement 1
*shows a dearth of points in the low expression/weak bias quadrant, as expected if ability to detect bias increases with expression.*
Figure 4—figure supplement 1
*also shows a weird dearth of points in the low expression/*high *bias quadrant. Explain? (Is that just the fact that the Bayesian estimation of parental bias is pulled back toward 0.5 by the prior, in cases of very low expression with little data?)*.

We thank reviewer for bringing our attention to this sensitive point. We realize our sentence is misleading. Higher expression levels do make it is easier to detect parental biases (as are more polymorphic sites that phase the reads, as well as lower redundancy between the different isoforms). Indeed, inference of parental biases at the range of very low expression levels (left parts of Figure 1—figure supplement 1) is significantly affected. The location of the parental biases at that range of expression levels reflects the low identifiability – MMSEQ reports high uncertainty in the allele-specific expression levels of these transcripts and our model propagates this uncertainty. As a result, the posterior distribution of the parental bias (which prior is at 0.5) thus has higher variance (at the most extreme case the posterior distribution will cover the entire 0.5-1 range), and the mean of that posterior distribution is thus likely to be close to 0.75. This trend is also visible when we contrast the parental bias inferred by RNA-seq and pyrosequencing versus expression levels (newly added Figure 1—figure supplement 1). Based on this figure we defined a more stringent expression level cutoff, where the discrepancy between the parental biases inferred by RNA-seq and pyrosequencing drops dramatically. At this expression level cutoff the dearth of imprinted genes in the left part of Figure 1—figure supplement 1 are removed, but our remaining analyses that depend on the RNA-seq data are negligibly affected.

*In the subsection “Transcriptome-Wide Analysis of Genomic Imprinting in the Cerebellum”, the sentence: “In addition to 74 imprinted genes previously validated… our analysis detects 41 novel imprinted genes… increasing the total number of reported imprinted genes in the mouse by 26%”. Unclear how the 26% number is arrived at; as with the comment about true positive rate, I don't think you've stated what you think the total number of reported imprinted genes is. This needs to be stated clearly, especially since*
[35]
*reported >1000, but was overestimating*.

We thank the reviewer for pointing this out. We compiled a list of all known imprinted genes up to our study (those that were reported as imprinting candidates and independently validated in addition). This list amounted in 138 known imprinted genes, which we increase to 179 (a ∼30% increase, which is actually slightly higher than we had in the submitted version, since the imprinting status of some genes that we counted as known is in fact highly questionable). This is now referred to in the subsection “Transcriptome-Wide Profile of Parent-of-Origin Expression in the Mouse Cerebellum”.

*I don't think the content of the paper has much to do with the “functional significance” of parental bias, nor does it have much to do with its “regulation” except in the descriptive sense that you've observed differences. The Title of the manuscript could just be “Parent of Origin Allelic Expression in the Brain”*.

We thank the reviewer for his suggestion and changed the Title to: “Quantitative and Functional Interrogation of Parent-of-Origin Allelic Axpression Biases in the Brain.” We believe that the phenotype observed in the paternal deletion of the BclX gene has significant functional implications for the understanding of the role of genomic imprinting in the brain.